# AuctionNet: A Novel Benchmark for Decision-Making in Large-Scale Games

**Kefan Su[1,2]\*, Yusen Huo[2], Zhilin Zhang[2], Shuai Dou[2], Chuan Yu[2], Jian Xu[2]†,**
**Zongqing Lu[1]† , Bo Zheng[2]**
[1]School of Computer Science, Peking University
[2]Alibaba Group
[1]{sukefan,zongqing.lu}@pku.edu.cn
[2] {huoyusen.huoyusen,zhangzhilin.pt,doushuai.ds,
yuchuan.yc,xiyu.xj,bozheng}@alibaba-inc.com

## Abstract

Decision-making in large-scale games is an essential research area in artificial intelligence (AI) with significant real-world impact. However, the limited access to realistic large-scale game environments has hindered research progress in this area. In this paper, we present AuctionNet, a benchmark for bid decision-making in large-scale ad auctions derived from a real-world online advertising platform. AuctionNet is composed of three parts: an ad auction environment, a pre-generated dataset based on the environment, and performance evaluations of several baseline bid decision-making algorithms. More specifically, the environment effectively replicates the integrity and complexity of real-world ad auctions through the interaction of several modules: the ad opportunity generation module employs deep generative networks to bridge the gap between simulated and real-world data while mitigating the risk of sensitive data exposure; the bidding module implements diverse auto-bidding agents trained with different decision-making algorithms; and the auction module is anchored in the classic Generalized Second Price (GSP) auction but also allows for customization of auction mechanisms as needed. To facilitate research and provide insights into the environment, we have also pre-generated a substantial dataset based on the environment. The dataset contains 10 million ad opportunities, 48 diverse auto-bidding agents, and over 500 million auction records. Performance evaluations of baseline algorithms such as linear programming, reinforcement learning, and generative models for bid decision-making are also presented as a part of AuctionNet. AuctionNet has powered the NeurIPS 2024 Auto-Bidding in Large-Scale Auctions competition, providing competition environments for over 1,500 teams. We believe that AuctionNet is applicable not only to research on bid decision-making in ad auctions but also to the general area of decision-making in large-scale games. Code[3]: https://github.com/alimama-tech/AuctionNet.

## 1 Introduction

Decision-making in large-scale games is a fundamental area of research in artificial intelligence. Agents in a large-scale game need to make strategic decisions to fulfill their objectives under certain constraints in a competitive environment. The research advances in this area have a profound impact on a broad range of real-world applications [13, 34, 35, 37]. Online advertising, with a market size of

---

\*This work is done during internship at Alibaba Group.

†Corresponding author.

[3]Alibaba Group retains full ownership rights to this benchmark.

38th Conference on Neural Information Processing Systems (NeurIPS 2024) Track on Datasets and Benchmarks.

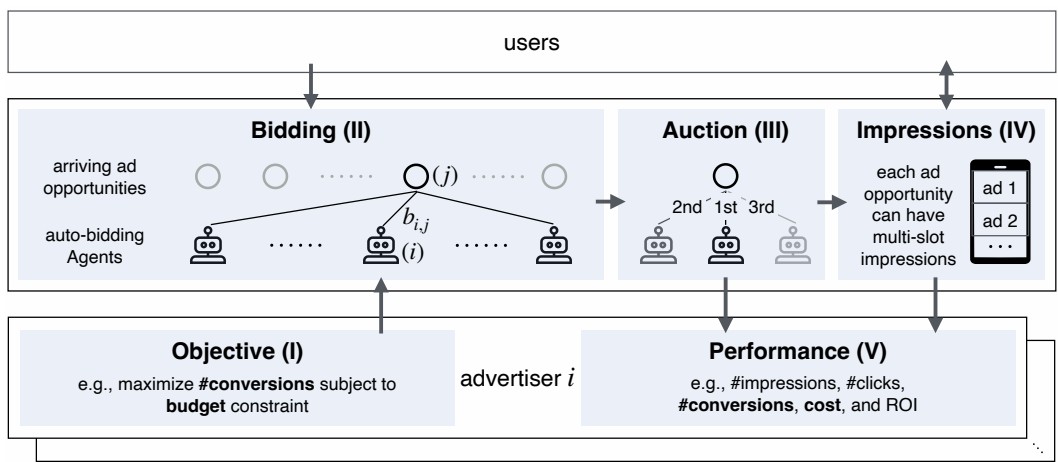

Figure 1: Overview of typical large-scale online advertising platform. Numbers 1 through 5 illustrate how an auto-bidding agent helps advertiser $i$ optimize performance. For each advertiser's unique objective (I), auto-bidding agent make bid decision-making (II) for continuously arriving ad opportunities, and compete against each other in the ad auction (III). Then, each agent may win some impressions (IV), which may be exposed to users and potentially result in conversions. Finally, the agents' performance (V) will be reported to advertisers.

more than \$600 billion in 2023, is perhaps one of the most representative applications that calls for sophisticated decision-making solutions in large-scale games. More specifically, as shown in Figure 1, a significant part of online advertising is based on real-time bidding (RTB), a process in which advertising inventory is bought and sold in real-time ad auctions. The auto-bidding agents strategically bid for impressions on behalf of the advertisers across a large number of continuously arriving ad opportunities to maximize performance, subject to certain constraints such as return-on-investment (ROI) [28].

Bid decision-making in large-scale ad auctions is a concrete example of decision-making in large-scale games. However, researchers usually only have limited access to realistic large-scale ad auction environments, hindering the research proccess in this area. Although a few existing works provide certain environments, there remains a considerable gap between these environments and the real-world environments. For instance, AuctionGym [18] overlooks changes in advertiser budgets across multiple auction rounds, while AdCraft [11] models competing bidders by sampling from a parameterized distribution, an approach that falls short of fully capturing the essence of the multi-agent dynamics inherent to this problem.

In this paper, we present AuctionNet, a benchmark for bid decision-making in large-scale ad auctions derived from a real-world online advertising platform. AuctionNet is composed of three parts: an ad auction environment, a pre-generated dataset based on the environment, and performance evaluations of a couple of baseline bid decision-making algorithms. More specifically, the environment effectively replicates the integrity and complexity of real-world ad auctions with the interaction of several modules: the ad opportunity generation module employs deep generative networks to bridge the gap between simulated and real-world data while mitigating the risk of sensitive data exposure; the bidding module implements diverse auto-bidding agents trained with different decision-making algorithms; and the auction module is anchored in the classic and popular Generalized Second Price (GSP) [9, 23, 7] auction but also allows customization of auction mechanisms as needed. To facilitate research and provide insights into the game environment, we also pre-generated a substantial dataset based on the environment. The dataset contains 10 million ad opportunities, 48 diverse auto-bidding agents, and over 500 million auction records. Performance evaluations of baseline algorithms such as linear programming, reinforcement learning, and generative models for bid decision-making are also presented as a part of AuctionNet.

We believe that AuctionNet is applicable not only to research on bid decision-making algorithms in ad auctions but also to the general area of decision-making in large-scale games. It can also benefit

researchers in a broader range of areas such as reinforcement learning, generative models, operational research, and mechanism design.

## 2 The Decision-Making Problem Concerned

In this paper, we are concerned with the auto-bidding problem in ad auctions. We use a Partially Observable Stochastic Game (POSG)[14] to formulate the problem. A POSG $\mathcal{M}$ can be represented as a tuple $\mathcal{M} = \{S, A, P, \boldsymbol{r}, \gamma, Z, O, I, T\}$, where $I = \{1, 2, \cdots, n\}$ is the set of all the agents, $T$ is the horizon, $i.e.$, the number of time steps in one episode, $S$ is the state space and $A$ is the action space, $P(\cdot|s, a) : S \times A \to \Delta(S)$ is the transition probability, $\gamma$ is the discount factor, $Z$ is the observation space, $O(s, i) : S \times I \to Z$ is the mapping from state to observation for each agent $i$, $\boldsymbol{r} = r_1 \times r_2 \times \cdots \times r_n$ is the joint reward function of all the agents, and $r_i(s, \boldsymbol{a}) : S \times A \to \mathbb{R}$ is the individual reward function for each agent $i$, where $\boldsymbol{a} = (a_1, a_2, \cdots, a_n) \in A = A_1 \times A_2 \times \cdots \times A_n$ is the joint action of all the agents.

Specifically, the interaction in one time step is as follows: The state $s = (\boldsymbol{\omega}, \boldsymbol{u}, \boldsymbol{q}, \boldsymbol{v})$ consists of budgets $\boldsymbol{\omega}$, ad opportunity features $\boldsymbol{u}$, advertiser features $\boldsymbol{q}$ such as industry category, corresponding value matrix $\boldsymbol{v} = \{v_{ij}\}$, where $v_{ij}$ is the value of ad opportunity $j$ for agent $i$. Agent $i$'s observation $o_i = (\omega_i, \boldsymbol{u}_i, q_i, \boldsymbol{v}_i) \in Z$ contains only part of the information in state $s$, $i.e.$, agent $i$ may not know the budgets of other agents. A convention in the auto-bidding area [3] proves that the optimal bid is proportional to the ad opportunity value. Following this convention, the action of agent $i$ is a coefficient $\alpha_i$, and the bids of agent $i$ for all the ad opportunities of this time step are $\boldsymbol{b}_i = (b_{i1}, b_{i2}, \cdots, b_{im}) = (\alpha_i v_{i1}, \alpha_i v_{i2}, \cdots, \alpha_i v_{im})$, where $m$ is the number of ad opportunities within this time step. Given the bids of all the agents, determined by the auction mechanism, agent $i$ will receive the auction result $\boldsymbol{x}_i = (x_{i1}, x_{i2}, \cdots, x_{im})$, where $x_{ij} = 1$ if and only if agent $i$ wins opportunity $j$. Agents will only receive rewards and incur costs from the winning impressions, $i.e.$, reward $r_i(s, \boldsymbol{a}) = \sum_{j=1}^{m} x_{ij} v_{ij}$ and budget for the next time step $\omega_i' = \omega_i - \sum_{j=1}^{m} x_{ij} c_{ij}$, where $c_{ij}$ is the cost of impression $j$ for agent $i$.

Taking a typical auto-bidding scenario as an example, given the definition above, the optimization objective from the perspective of agent $i$ is as follows:

$$\underset{\{\alpha_i^t\}}{\text{maximize}} \sum_{t=1}^{T} \langle \boldsymbol{x}_i^t, \boldsymbol{v}_i^t \rangle \quad \text{s. t.} \sum_{t=1}^{T} \langle \boldsymbol{x}_i^t, \boldsymbol{c}_i^t \rangle \leq \omega_i, \tag{1}$$

where $\boldsymbol{x}_i^t = (x_{i1}^t, x_{i2}^t, \cdots, x_{im}^t)$, $\boldsymbol{v}_i^t = (v_{i1}^t, v_{i2}^t, \cdots, v_{im}^t)$, $\boldsymbol{c}_i^t = (c_{i1}^t, c_{i2}^t, \cdots, c_{im}^t)$, $\omega_i$ is the budget of agent $i$, and $\langle \cdot \rangle$ denotes the inner product. As for the implementation, we know from our problem formulation that $r_i(s_t, \boldsymbol{a}_t) = \langle \boldsymbol{x}_i^t, \boldsymbol{v}_i^t \rangle$, so the objective in the optimization formulation is the same as $\sum_{t=1}^{T} r_i(s_t, \boldsymbol{a}_t)$. For more complex scenarios, we can add the CPA constraint to ensure effective utilization of the budget. More details on these CPA-constrained problems are included in Appendix E. The decision-making formulation above can be easily extended to various real-world scenarios.

## 3 Ad Auction Environment

To comprehensively demonstrate large-scale games from real-world online advertising platforms, we have developed an ad auction environment. To standardize the auto-bidding process, we divide ad opportunities within a period into $T$ decision time steps. Given the objective, the auto-bidding agent sequentially bids at each step, using the results from step $t$ and prior historical information to refine its strategy for step $t + 1$. This design philosophy enables agents to continuously optimize their bidding strategies in order to adapt to the changing environment. Within each step, all ad opportunities are executed independently and in parallel. At the end of the period, the environment provides the final performance for the agent.

The environment effectively replicates the integrity and complexity of real-world ad auctions through the interaction of several modules: the ad opportunity generation module, the bidding module, and the auction module. To better simulate large-scale auctions in reality, a substantial number of ad opportunities are fed into the environment and configured with dozens of bidding agents. These ad opportunities are generated using deep generative networks to reduce the gap between the simulation

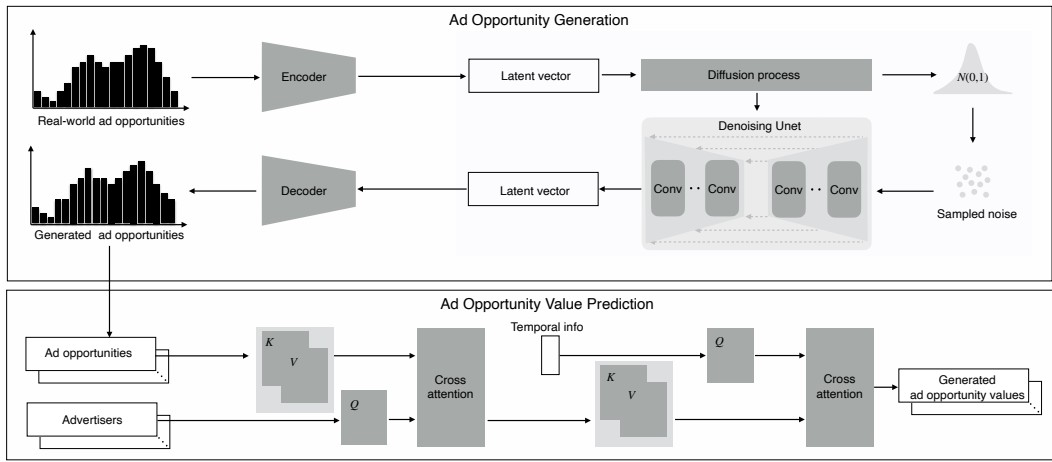

Figure 2: Overview of the pipeline of the ad opportunity generation network. The generation process consists of two stages. In the first stage, ad opportunity features are generated through a latent diffusion model. In the second stage, the value prediction for the generated ad opportunity features is performed, incorporating both the time feature and the advertiser feature. Moreover, the volume of ad opportunities fluctuates over time, mirroring that of real-world online advertising platforms.

environment and reality while avoiding the risks of sensitive data exposure. The agents are equipped with diverse and sophisticated auto-bidding algorithms.

### 3.1 The Ad Opportunity Generation Module

The target of the ad opportunity generation module is to generate diverse ad opportunities similar to real online advertising data with deep generative networks, as shown in Figure 2. We aimed to adopt the diffusion model to generate ad opportunity but encountered difficulties with the denoising operation, which can yield unreasonable outputs. Therefore, we followed the approach of the Latent Diffusion Model (LDM) [25] to generate ad opportunity. LDM adds noise and performs denoising in the latent space using a diffusion model, and then generates data from the latent space with an encoder and decoder. Specifically, LDM maps the ad opportunity feature $u$ to a latent vector $y$ with the encoder and reconstructs this feature with the decoder during training. For generation, LDM samples a random latent vector from a normal distribution and then generates an ad opportunity feature based on this vector. Let $U \subset \mathbb{R}^d$ be the space of ad opportunity feature data $(u_1, u_2, \cdots, u_K)$, where $d$ is the dimension of the original data and $K$ is the number of ad opportunities. Let $Y \subset \mathbb{R}^{d'}$ be the latent space ($d' < d$). The encoder and decoder are represented as $g_\phi$ and $h_\psi$, respectively, where $\phi$ and $\psi$ are the parameters. The function of the encoder $g_\phi$ is to obtain a latent representation of the original data as $g_\phi(u_k) = (\mu_k, \sigma_k)$, where $y_k \sim \mathcal{N}(\mu_k, \sigma_k^2)$ and $y_k \in Y$ is the latent representation. In practice, the reparameterization trick [20] is applied to ensure that this operation is differentiable during backpropagation.

Given the latent representation $y_k$, the decoder is responsible for reconstructing the original data from $y_k$, i.e., $h_\psi(y_k) = \tilde{u}_k \in U$. In addition to the reconstruction, the latent distribution $\mathcal{N}(\mu_k, \sigma_k^2)$ is expected to approximate the standard Gaussian distribution $\mathcal{N}(0, 1)$. Therefore, we have the following loss function for the encoder and decoder:

$$\mathcal{L}_{\text{recons}} = \frac{1}{K} \sum_{k=1}^{K} \|u_k - h_\psi(y_k)\|_2^2, \quad \mathcal{L}_{\text{reg}} = \frac{1}{K} \sum_{k=1}^{K} D_{\text{KL}} \left( \mathcal{N}(\mu_k, \sigma_k^2) \big\| \mathcal{N}(0, 1) \right),$$

where $\mathcal{L}_{\text{recons}}$ is the reconstruction loss and $\mathcal{L}_{\text{reg}}$ is the regularization loss for the latent distribution.

Different from the original idea of VAE [20], where the latent variable $y \in Y$ is sampled from $\mathcal{N}(0, 1)$ in the generation process, LDM uses a diffusion model in the latent space to generate the latent variable. In general, the idea behind the diffusion model is to add Gaussian noise to the original data to obtain variables that follow $\mathcal{N}(0, 1)$ and to denoise from $\mathcal{N}(0, 1)$ for generation. Given a

latent variable $y$, we denote its noisy version after $p$ iterations as $y_p$. The diffusion model includes a network to predict noise $\epsilon_\theta(y_p, p)$, and the loss function can be represented as

$$\mathcal{L}_{LDM} = \frac{1}{K} \sum_{k=1}^{K} \left\| \epsilon_k - \epsilon_\theta(y_{k,p_k}, p_k) \right\|_2^2,$$

where $\epsilon_k \sim \mathcal{N}(0,1)$, $y_k$ is the latent embedding of $u_k$, and $p_k$ is uniformly sampled from the set $\{1, 2, \cdots, p_{\max}\}$. The network $\epsilon_\theta(y_p, p)$ is the only learnable component in the diffusion model, which enables the process of adding noise and denoising through basic operations.

As for the generation process, a latent variable $\bar{y}$ is sampled from $\mathcal{N}(0,1)$, and $\tilde{y}$ is obtained through $p_{\max}$ denoising steps from $\bar{y}$ using the noise prediction network $\epsilon_\theta$. Finally, the decoder generates an ad opportunity feature based on $\tilde{y}$ as $\tilde{u} = h_\psi(\tilde{y})$.

Given an ad opportunity feature $u_k$, we also need to determine the value of this ad opportunity combined with the category information of the corresponding advertiser $q_k$ and the time information $u_k^{\text{time}}$, where $q_k$ is the advertiser information in the real-world data associated with $u_k$. We use Multi-head Attention (MHA) [31] as the network architecture for information integration. Let $v_\xi$ represent the value prediction module, and $v_\xi(u_k, q_k, u_k^{\text{time}})$ denote the predicted value of the ad opportunity feature $u_k$ for a specific advertiser at a specific time step. The loss of the value prediction model is shown below:

$$\mathcal{L}_{\text{pred}} = \frac{1}{K} \sum_{k=1}^{K} \left\| v_k - v_\xi(u_k, q_k, u_k^{\text{time}}) \right\|_2^2,$$

where $v_k$ is the true value of the ad opportunity in the record associated with $u_k$.

### 3.2 The Bidding Module

The bidding module replicates the dynamic competition between advertisers, each of whom has distinct advertising objectives and utilizes a separate auto-bidding agent, while remaining unaware of their competitors' strategies. Researchers can control a subset of the agents in the environment, while other agents remain uncontrollable, thereby better reflecting the complex and dynamic game in real-world online advertising.

Several algorithms in the auto-bidding area have been implemented as baselines, including the PID Controller [36], Online LP [15], IQL [21], Behavior Cloning [30], and Decision Transformer [8]. This facilitates researchers who are interested in quickly starting up and evaluating these baselines in a unified environment.

### 3.3 The Auction Module

The task of the auction module is to determine the winner and the winning price given all the bids from agents for ad opportunities. The costs for agents will vary depending on the different auction rules. The most commonly discussed auction rule is the Generalized Second-Price (GSP) Auction, which stipulates that the winner pays a cost slightly higher than the second-highest bid rather than the highest bid. The auction module internally supports several popular auction rules, including GSP, for the convenience of researchers. Additionally, researchers can design specific auction rules tailored to their purposes using the interface of the auction module.

Additionally, the property of multiple slots has been implemented in the environment. Multiple slots arise from applications in the industry, meaning that a single ad opportunity may have multiple ad slots for display. A slot with a higher exposure rate is more valuable to advertisers. Suppose the number of slots is $l$, then the auction module will allocate $l$ slots to the top $l$ bidders, and these bidders will receive different values according to the varying exposure rates of the slots. In summary, the multiple slots feature increases the complexity of the optimal bidding strategy, as the exposure rate serves as a discount factor for both cost and value.

### 3.4 API

The code of the environment is implemented in Python. The environment API is similar to OpenAI Gym[5], so the construction and interactions of the environment may be familiar to related researchers. We included an example code as follows:

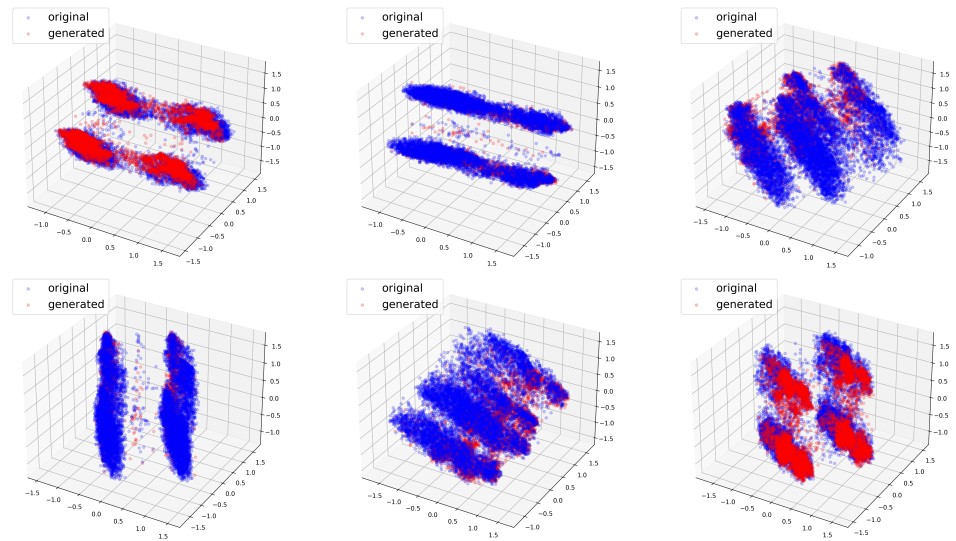

Figure 3: The 3D PCA results of 100K generated data and 100K real-world data.

```python
from AuctionNet import Controller
# Load player agent
bidding_controller = Controller(player_agent=player_agent)
# Init other competing agents
agents = bidding_controller.agents
# Init auction module
envs = bidding_controller.biddingEnv
# Generate ad opportunities
ad_opportunities = bidding_controller.adOpportunityGenerator.generate()
# Init the budget and reward of each agent
rewards = np.zeros(shape=(len(agents)))
costs =  np.zeros(shape=(num_agents))
for episode in range(num_episode):
    for tick_index in range(num_tick):
        # load ad opportunities
        tick_ad_opportunities = ad_opportunities[episode][tick_index]
        # Collect bids from each agent
        bids = []
        for agent in agents:
            bids.append(agent.bidding())
        # Simulate bidding process
        auction_res = envs.simulate_ad_bidding(tick_ad_opportunities, bids)
        # Aggregate bidding results
        rewards+=auction_res["reward"]
        costs+=auction_res["cost"]
```

## 4 Pre-Generated Dataset Based on the Environment

In this section, we first verify whether the ad opportunity generation module can generate ad opportunity features similar to those in real-world data. Next, we briefly introduce and analyze the dataset generated from the AuctionNet environment.

### 4.1 Verification of the Ad Opportunity Generation Module

In order to better demonstrate that the generated data can reflect the properties of real-world data, the effectiveness of the ad opportunity generation module itself was verified. The ad opportunity

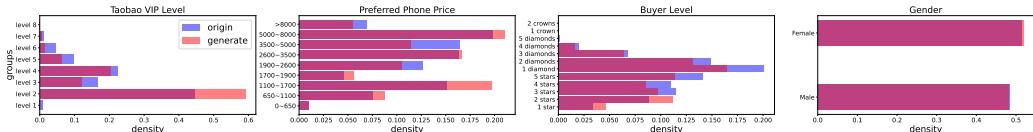

Figure 4: The distribution of identity information including the Taobao VIP level, the preferred phone price, the buyer level, and the gender in 100K generated data and 100K real-world data.

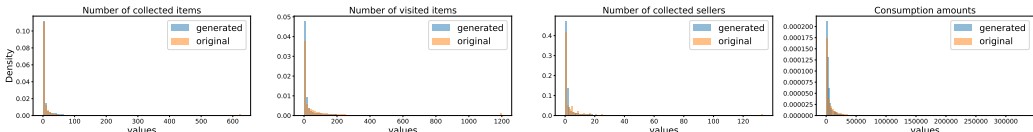

Figure 5: The distribution of consumption behavior information including the number of collected items, the number of visited items, the number of collected sellers, and the consumption amounts in 100K generated data and 100K real-world data.

generation module comprises two components: a feature generation model and a value prediction model. Experiments were conducted to verify the effectiveness of these models.

We randomly sample 100K real-world online advertising data points to compare with 100K generated data points. The details of the generated data can be found in Appendix D. First, we perform PCA [19] to visualize the similarity between the real-world and generated data. The 3D PCA results are illustrated in Figure 3. For better presentation, we use six different views in the 3D space. We observe that the generated data overlap with the original data in the 3D space. Moreover, the generated data points form four main separate clusters in the 3D space, similar to the real-world data points. These visualization results demonstrate that the generated data generally resemble the real-world data.

To further compare these two datasets, we study the value distributions of identity information and consumption behavior information in both datasets. The empirical results are included in Figure 4 and Figure 5. The feature vector contains over 20 fields, as described in Appendix D, so we only select a subset of these fields for our experiments. Regarding identity information, the generated value distributions are similar to the real-world value distributions overall, although biases exist for certain terms, such as 'level 7' for the Taobao VIP Level. Distributions with more categories are more challenging to match, while the gender distributions are nearly identical in both datasets. For consumption behavior information, we observe that the distributions in the selected fields share a strong resemblance and exhibit long-tail characteristics. A long-tail distribution indicates that most users do not engage in frequent consumption, and users with a high volume of consumption behavior are rare. This phenomenon aligns with our experience in online advertising.

We investigate whether the generated data can capture the connections between different fields. Based on the observation that users with higher VIP levels typically exhibit a higher volume of consumption behavior, we examine the connection between the Taobao VIP level and consumption behavior. We select four consumption behavior fields. The mean values of these fields across different VIP levels are shown in Figure 6. We find that the overall monotonically increasing trend is captured by the generated data, although biases exist in the specific values. Moreover, the drop in values from 'level 7' to 'level 8' is also captured by the generated data in three out of the four fields, except for the consumption amount. The rarity of 'level 8' data points may be the reason why the generative model is unable to distinguish different trends for different fields.

In real-world online advertising, the metrics for bidding strategy evaluation are Click-Through Rate (CTR) and Conversion Rate (CVR). Bidding strategies make decisions based on the predicted CTR (pCTR) and predicted CVR (pCVR), which are the estimated values of CTR and CVR, respectively. For simplicity, in this environment, we assume that the estimations are accurate and define the value as $\text{value} = \text{pCTR} \cdot \text{pCVR}$. Our value prediction model learns to predict pCTR and pCVR and subsequently calculates the value. We predict the pCTR, pCVR, and value for 100K real-world data points and compare these predictions with the real-world ground truth.

We hope that the value prediction model can capture the value variation over changes in category and time. The means of predicted pCTR, pCVR, and values across different categories and time steps,

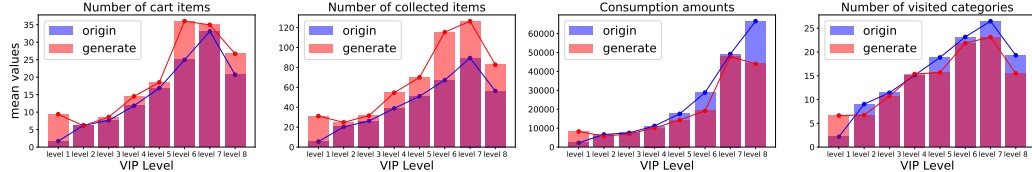

Figure 6: The mean values of consumption behavior information including the number of cart items, the number of collected items, the consumption amounts, and the number of visited categories in different VIP levels in 100K generated data and 100K real-world data.

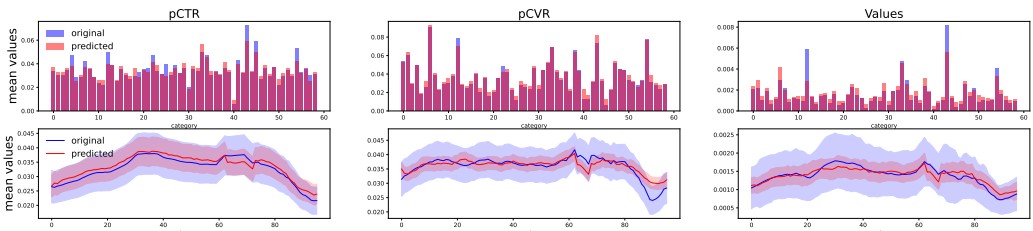

Figure 7: The means of the predicted pCTR, pCVR, and value in different categories and time steps compared with the ground truth. The shaded areas are related to the standard deviation.

compared with the ground truth, are illustrated in Figure 7. The empirical results show that, in general, the variation trends in predictions over changes in category and time are similar to the ground truth.

To present the results more intuitively, we provide additional quantitative results. We compare the mean squared error (MSE) between the generated and original distributions with the standard deviation of the original distribution. The quantitative results are shown in Table 1. It can be observed that the MSEs are all smaller than the original standard deviations (original_stds), indicating that our prediction model can capture the patterns of value variation and is accurate.

## 4.2 Pre-Generated Dataset

The dataset is derived from game data generated within the environment, where numerous auto-bidding agents compete against each other. We have pre-generated large-scale game data to assist researchers in gaining deeper insights into the auction ecosystem. This data can be used to model the environment and to train the auto-bidding agents effectively.

The dataset contains 10 million ad opportunities, including 21 advertising episodes. Each episode contains more than 500,000 ad opportunities, divided

Table 1: The comparison of the MSE between the generated and original distribution with the standard deviation of the original distribution.

|  | original_std | MSE |
|---|---|---|
| pCVR_category | 0.0685 | **0.0341** |
| pCTR_category | 0.0517 | **0.0280** |
| value_category | 0.00573 | **0.00496** |
| pCVR_time | 0.0637 | **0.0313** |
| pCTR_time | 0.0590 | **0.0259** |
| value_time | 0.00625 | **0.00176** |

into 48 steps. Each opportunity includes the top 48 agents[4] with the highest bids. The dataset comprises over 500 million records, totaling 80 GB in size. Each record includes information such as the predicted value, bid, auction, and impression results, among other details. The specific data format and data samples of the dataset are included in Appendix C.

We have conducted an analysis of the AuctionNet Dataset to provide some insights. We first investigate the variation of impression values over time within a single day. We selected five categories from the AuctionNet Dataset and denote them as Category 1, Category 2, and so on. As in Figure 8, the impression values of different categories exhibit distinct patterns of variation. Given the budget constraint, agents should consider the variation in impression values over time to bid for appropriate impressions at the optimal times. Furthermore, we examine the relations between the values of different categories. The relations between Category 1 and other categories are illustrated in Figure 9.

---

[4]Real-world data show that 48 agents can ensure competitive pressure for auto-bidding agent training.

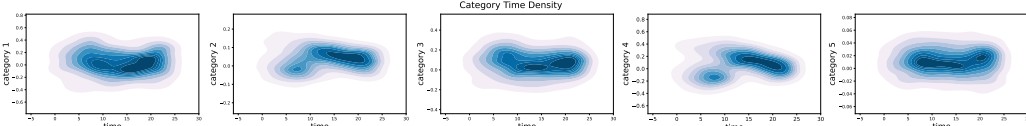

Figure 8: The joint value distribution between different categories and time in the dataset.

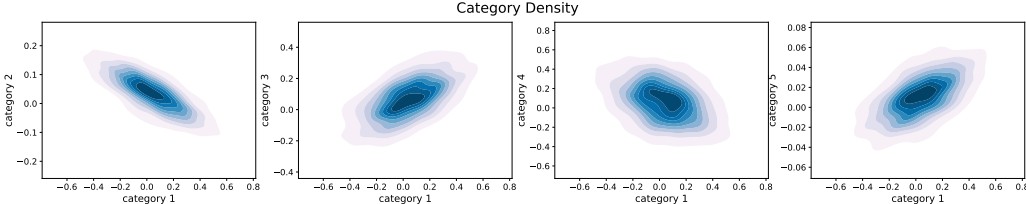

Figure 9: The joint value distribution between Category 1 and other categories in the dataset.

The impression values of Category 1 and Category 3 are positively correlated, indicating that the corresponding advertisers are competitors for similar ad opportunities. Therefore, considering the preferences of other agents may be beneficial for developing better bidding strategies. The full datasheet of the dataset is included in Appendix B.

## 5 Performance Evaluations of Baseline Algorithms

In this section, we evaluate the performance of baseline algorithms, such as linear programming, reinforcement learning, and generative models. It is important to note that we used the original algorithms from the papers and did not perform any special optimization on the methods specifically for the auto-bidding tasks. We provide a brief introduction to these baselines. The idea of the PID Controller is straightforward: it uses three parameters, $\lambda_P$, $\lambda_I$, and $\lambda_D$, for Proportional Control, Integral Control, and Derivative Control, respectively. In this baseline, the PID Controller is employed to control the cost or bids of agents. Online LP utilizes linear programming for the auto-bidding problem. At each time step, Online LP solves a knapsack problem using a greedy algorithm. IQL is an offline RL algorithm. The core idea behind IQL is to evaluate the offline Q-function only on actions that appeared in the offline data, thereby avoiding overestimation in out-of-distribution data. Behavior Cloning (BC) is a supervised learning algorithm that uses expert trajectories. The agent's policy is learned by predicting the expert's actions in the state of given trajectories. Decision Transformer (DT) leverages the capabilities of the Transformer model[31] for sequential decision-making. DT treats the trajectories in a MDP as sequences and predicts actions based on previous transitions. More generative models such as AIGB [12] will also be integrated into baseline algorithms in the future. To better illustrate the performances, we add a heuristic method, Abid, to the experiments. Abid means the agent will give a fixed bid rate for all impressions. Its performance can be seen as a reference in comparison. More details of the evaluation can be found in Appendix A.

The empirical results are included in Figure 10. For better illustration, we normalize the performances of all baselines by the mean episode reward of the heuristic baseline Abid. Therefore, the mean relative performance of Abid is $1.0$ in the basic task. Online LP achieves the best performance, possibly because it is relatively robust and does not require special adaptation for auto-bidding tasks to achieve good results. Although methods like IQL and BC perform not as well as Online LP, we observe that proposing optimized solution [12, 22]can significantly optimize the performance, proving that such methods have great potential for optimization. In addition, the drop in rewards observed for all baselines during the target CPA task is due to the CPA penalty for exceeding constraints in (4).

## 6 Applications

AuctionNet has powered the the NeurIPS 2024 competition "Auto-bidding in Large-Scale Auctions" [1]. The competition addressed the critical issue of making high-frequency bid decision-making in uncertain and competitive environments and attracted more than 1,500 teams

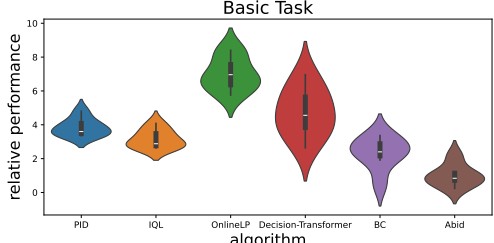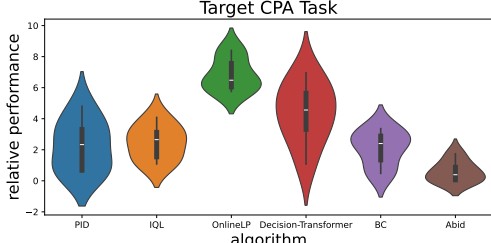

Figure 10: The empirical results of baseline algorithms on the basic task and Target CPA task.

from around the world to participate, lasting for 4 months. The ad auction environment, dataset, and baseline bid decision-making algorithms used in the competition are derived from this benchmark. The ad auction environment provided nearly ten thousand evaluations for the competition, offering participants accurate and fair performance assessments. The dataset and baseline algorithms allowed participants to quickly start the task and stimulated their creativity, leading to more diverse and innovative solutions, thus driving technological development in this area.

# 7 Related Work

Simulation environments have been widely applied in decision-making research and have successfully promoted the development of related studies [6, 24, 32, 27, 29]. However, simulation environments for real-world online advertising platforms are relatively scarce in the bid decision-making field. AuctionGym [18] models the bidding problem as a contextual bandit problem [2], where the advertiser decides the bidding value given the information of the ad opportunity as context. The contextual bandit has only one time step per episode, meaning that AuctionGym does not consider budget constraints in auto-bidding. Moreover, AuctionGym describes the auto-bidding problem from a single-agent perspective and ignores the influence of other agents. AdCraft [11] is a simulation environment for the bidding problem in Search Engine Marketing (SEM). Although AdCraft explicitly models the influences of other agents, these agents' policies are sampled from parameterized distributions, which cannot fully reflect the multi-agent nature of this problem. Despite the points discussed above, these existing simulation environments lack data-driven methods for modeling real-world online advertising platforms.

# 8 Conclusion and Limitations

We present AuctionNet, a benchmark for bid decision-making in large-scale ad auctions derived from a real-world online advertising platform. AuctionNet consists of three components: an ad auction environment augmented with verified deep generative networks, a pre-generated dataset based on this environment, and performance evaluations of several baseline bid decision-making algorithms. The AuctionNet not only provides researchers with the opportunity to study auto-bidding algorithms in large-scale auctions, but also helps researchers and practitioners in game theory, reinforcement learning, generative models, operations optimization, and other fields to solve a wide range of decision-making research problems. Regarding limitations, while the generated data in the AuctionNet environment and the real-world data are similar in general, there are biases in some details, and the performance of the generative model can be improved.

# 9 Acknowledgments

This work was supported in parts by NSFC under grants 62450001 and 62476008 and Alibaba Group through Alibaba Innovative Research Program. The authors would like to thank the anonymous reviewers for their valuable comments and advice.

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

# A Evaluation Details

There are 48 agents of 7 types in our experiments and each type corresponds to one algorithm. We test 7 rounds where we permute the order of agents in each round. Therefore, agents will represent different advertisers with different budgets in different rounds. We choose the best agent as the representative of an algorithm if there are multiple agents of this algorithm. We use the average performances of the 7 rounds as the final performance of all the algorithms. We provide the model file of these agents and the evaluation code for reproduction.

# B Datasheet for AuctionNet

We present a datasheet[10] for the AuctionNet Dataset.

## B.1 Motivation

### For what purpose was the dataset created?

In general, learning from interactions with the real-world online advertising platforms is difficult and expensive, so offline RL algorithms are more popular in auto-bidding. Therefore, we build the Auction Dataset to facilitate offline training of users. Moreover, the Auction Dataset will also be provided to the participants of the competition we will hold in the future.

### Who created the dataset?

The dataset was created by the authors of this paper. The dataset was not created on the behalf of any entity.

### Who funded the creation of the dataset?

Alibaba Group funds the creation of the AuctionNet Dataset.

## B.2 Composition

### What do the instances that comprise the dataset represent (e.g., documents, photos, people, countries)?

The AuctionNet Dataset contains trajectories of diverse agents competing with each other. Please refer to Appendix C and Section 4.2 for more details.

### Is there a label or target associated with each instance?

The AuctionNet Dataset contains offline trajectories where the actions or bids of agents can be seen as labels for the time step.

### Is any information missing from individual instances?

Not to our knowledge.

### Are there recommended data splits (e.g., training, development/validation, testing)?

No.

### Are there any errors, sources of noise, or redundancies in the dataset?

The AuctionNet Dataset contains trajectories of diverse agents, some of these agents may not perform well. However, the tasks in the environment are still difficult for some algorithms and we think keeping agents diverse in the AuctionNet Dataset is beneficial.

### Do/did we do any data cleaning on the dataset?

We did not. All data is presented exactly as collected.

## B.3 Collection Process

### How was the data associated with each instance acquired?

The AuctionNet Dataset is collected from the interactions of baseline agents in the environment.

**Who was involved in the data collection process and how were they compensated?**

The data collection process is done by the authors and not involve with any crowdsource.

**Over what timeframe was the data collected?**

The AuctionNet Dataset was collected between March 2024 and May 2024.

## B.4   Uses

**Has the dataset been used for any tasks already?**

No.

**Is there a repository that links to any or all papers or systems that use the dataset?**

No.

**Is there anything about the composition of the dataset or the way it was collected and preprocessed/cleaned/labeled that might impact future uses?**

We do not believe so since the AuctionNet Dataset consists of data generated by the interactions of baseline agents.

## B.5   Distribution

**Will the dataset be distributed to third parties?**

Yes, but the AuctionNet Dataset and environment are involved with a large competition we will hold in NeurIPS 2024, so we will not distribute them until the end of the competition considering competition fairness. However, we will open-source the AuctionNet Dataset as soon as possible.

**How will the dataset will be distributed (e.g., tarball on website, API, GitHub)? Does the dataset have a digital object identifier (DOI)?**

The AuctionNet Dataset will be distributed by a Github link after the end of the competition we will hold. The AuctionNet Dataset doesn't have a digital object identifier now.

All data is under the MIT license.

**Have any third parties imposed IP-based or other restrictions on the data associated with the instances?**

No.

**Do any export controls or other regulatory restrictions apply to the dataset or to individual instances?**

No.

## B.6   Maintenance

**Who will be supporting/hosting/maintaining the dataset?**

The authors of this paper will provide needed maintenance to the datasets.

**How can the owner/curator/manager of the dataset be contacted (e.g., email address)?**

Please email us at huoyusen.huoyusen@alibaba-inc.com.

**Is there an erratum?**

There is not and we believe generated features, predicted values, and trajectories in our datasets do not involve an erratum.

**Will the dataset be updated (e.g., to correct labeling errors, add new instances, delete instances)?**

Yes, but as we won't add extra data points, the update will be minimal.

## C  Data Format of AuctionNet Dataset

The specific data format of the AuctionNet Dataset is as follows:

(c1). deliveryPeriodIndex: The index of the current delivery period.

(c2). advertiserIndex: The unique identifier of the advertiser.

(c3). advertiserCategoryIndex: The index of the advertiser's category.

(c4). budget: The advertiser's budget for a period.

(c5). CPAConstraint: The CPA constraint of the advertiser.

(c6). timeStepIndex: The index of the current decision time step.

(c7). remainingBudget: The advertiser's remaining budget before the current step.

(c8). pvIndex: The index of the ad opportunity.

(c9). pValue: The conversion probability when the ad is exposed to the user.

(c10). pValueSigma: The variance of predicted probability.

(c11). bid: The agent's bid of the ad opportunity.

(c12). xi: The winning status of the agent of the ad opportunity.

(c13). adSlot: The won ad slot.

(c14). cost: The cost needs to be paid if the ad is exposed to the user.

(c15). isExposed: The indicator signifying whether the ad in the slot was displayed to the user.

(c16). conversionAction: The indicator signifying whether the conversion action has occurred.

(c17). leastWinningCost: The minimum cost to win the ad opportunity.

(c18). isEnd: The completion status of the advertising period.

Table 2 presents an ad opportunity involving the top five advertisers. The top three advertisers, numbered 31, 22, and 15, won the ad opportunity with the highest bids and were allocated to ad slots 1, 2, and 3, respectively. During this impression, slots 1 and 2 were exposed to the user, while slot 3 remained unexposed. Consequently, ads in slots 1 and 2 need to pay 0.2702 and 0.2154, respectively. Additionally, the user engaged in a conversion action with the ad in slot 2.

Table 2: Bidding, auction, and impression processes for each advertiser during the same opportunity.

| c1 | c2 | c3 | c4 | c5 | c6 | c7 | c8 | c9 | c10 | c11 | c12 | c13 | c14 | c15 | c16 | c17 | c18 |
|---|---|---|---|---|---|---|---|---|---|---|---|---|---|---|---|---|---|
| 1 | 31 | 2 | 6500.00 | 27.00 | 5 | 5962.49 | 101000 | 0.0103542 | 0.0021549 | 0.2845 | 1 | 1 | 0.2702 | 1 | 0 | 0.1832 | 0 |
| 1 | 22 | 6 | 7000.00 | 38.00 | 5 | 5988.25 | 101000 | 0.0070297 | 0.0005213 | 0.2702 | 1 | 2 | 0.2154 | 1 | 1 | 0.1832 | 0 |
| 1 | 15 | 7 | 7000.00 | 42.00 | 5 | 6132.52 | 101000 | 0.0051392 | 0.0004312 | 0.2154 | 1 | 3 | 0.1832 | 0 | 0 | 0.1832 | 0 |
| 1 | 39 | 3 | 6000.00 | 30.00 | 5 | 5443.27 | 101000 | 0.0062134 | 0.0007254 | 0.1832 | 0 | 0 | 0 | 0 | 0 | 0.1832 | 0 |
| 1 | 43 | 9 | 7500.00 | 25.00 | 5 | 6421.81 | 101000 | 0.0045392 | 0.0006215 | 0.1099 | 0 | 0 | 0 | 0 | 0 | 0.1832 | 0 |

Table 3 presents a data sample illustrating an advertiser's bidding process across time steps within a delivery period. The advertiser has a budget of 7500, a CPA constraint of 40, and belongs to industry category 6. Throughout different time steps, the advertiser engages in bidding for every available impression and obtains the corresponding results. During this period, the advertiser's remaining budget decreases correspondingly. Additionally, the advertiser adjusts their bidding strategy based on prior performance, although this adjustment will not be directly evident in the data.

## D  The Structure of Generated Data

**Structure of the feature vector.** The feature vector consists of several types of information including one-hot vectors, integers and float numbers. The specific data format of the feature vector is as follows:

(c1). idAgeLevel: Represents the age level of the user. The meanings of values: 0 for unknown, 1~8 for ages over 12, 18, 22, 25, 30, 35, 40 and 50 respectively. Data format: one-hot vector, dimension $[0, 9)$.

Table 3: An advertiser's bidding process across time steps.

| c1 | c2 | c3 | c4 | c5 | c6 | c7 | c8 | c9 | c10 | c11 | c12 | c13 | c14 | c15 | c16 | c17 | c18 |
|---|---|---|---|---|---|---|---|---|---|---|---|---|---|---|---|---|---|
| 3 | 48 | 6 | 7500.00 | 40.00 | 1 | 7500.00 | 1 | 0.0032157 | 0.0003567 | 0.1345 | 0 | 0 | 0 | 0 | 0 | 0.1628 | 0 |
| 3 | 48 | 6 | 7500.00 | 40.00 | 1 | 7500.00 | 2 | 0.0146256 | 0.0021352 | 0.5852 | 0 | 0 | 0 | 0 | 0 | 0.6421 | 0 |
| 3 | 48 | 6 | 7500.00 | 40.00 | 1 | 7500.00 | 3 | 0.0054324 | 0.0007631 | 0.1924 | 1 | 1 | 0.1673 | 1 | 1 | 0.1454 | 0 |
| 3 | 48 | 6 | 7500.00 | 40.00 | 1 | 7500.00 | 4 | 0.0073145 | 0.0006529 | 0.2786 | 0 | 0 | 0 | 0 | 0 | 0.2862 | 0 |
| | | | | | | | ... | | | | | | | | | | |
| 3 | 48 | 6 | 7500.00 | 40.00 | 2 | 7341.25 | 20901 | 0.0076453 | 0.0006579 | 0.2856 | 0 | 0 | 0 | 0 | 0 | 0.3245 | 0 |
| 3 | 48 | 6 | 7500.00 | 40.00 | 2 | 7341.25 | 20902 | 0.0139234 | 0.0012358 | 0.5629 | 1 | 2 | 0 | 0 | 0 | 0.6782 | 0 |
| 3 | 48 | 6 | 7500.00 | 40.00 | 2 | 7341.25 | 20903 | 0.0077212 | 0.0006579 | 0.3045 | 0 | 0 | 0 | 0 | 0 | 0.3122 | 0 |
| 3 | 48 | 6 | 7500.00 | 40.00 | 2 | 7341.25 | 20904 | 0.0021341 | 0.0001873 | 0.0926 | 0 | 0 | 0 | 0 | 0 | 0.1151 | 0 |
| | | | | | | | ... | | | | | | | | | | |
| 3 | 48 | 6 | 7500.00 | 40.00 | 43 | 0.00 | 895201 | 0.0065274 | 0.0005689 | 0.0000 | 0 | 0 | 0 | 0 | 0 | 0.1243 | 1 |
| 3 | 48 | 6 | 7500.00 | 40.00 | 43 | 0.00 | 895202 | 0.0032125 | 0.0002986 | 0.0000 | 0 | 0 | 0 | 0 | 0 | 0.2986 | 1 |
| 3 | 48 | 6 | 7500.00 | 40.00 | 43 | 0.00 | 895203 | 0.0112986 | 0.0013253 | 0.0000 | 0 | 0 | 0 | 0 | 0 | 0.0932 | 1 |
| 3 | 48 | 6 | 7500.00 | 40.00 | 43 | 0.00 | 895204 | 0.0051678 | 0.0006782 | 0.0000 | 0 | 0 | 0 | 0 | 0 | 0.1687 | 1 |

(c2). idGender: Represents the gender of the user. The meanings of values: 0,1 and 2 for unknown, Female and Male respectively. Data format: one-hot vector, dimension $[9, 12)$.

(c3). isForeign: Represents whether the user is foreign. The meanings of values: 0,1 and 2 for unknown, No and Yes respectively. Data format: one-hot vector, dimension $[12, 15)$.

(c4). cityLevel: Represents the level of the city where the user is living. The meanings of values: 0 for unknown, 1~6 for different city development levels in descending order. Data format: one-hot vector, dimension $[15, 22)$.

(c5). isCap: Represents whether the city the user living in is the capital. The meanings of values: 0 for No and 1 for Yes. Data format: one-hot vector, dimension $[22, 24)$.

(c6). buyerStarName: Represents the rating of the user as a buyer. The meanings of values: 0 for unknown, 1~5 for 1~5 stars respectively, 6~10 for 1~5 diamonds respectively, 11~15 for 1~5 crowns respectively, 16~20 for 1~5 golden crowns respectively, 21 for credit score $\leq 3$, and 22 for credit score $= 0$. In general, the order of these values' ratings is $22 < 21 < 1 < 2 < \cdots < 20$. Data format: one-hot vector, dimension $[24, 47)$.

(c7). tmLevel: Represents the VIP level of the user in Tmall. The meanings of values: 0 for unknown or no VIP level, 1~5 for VIP levels 1~5 respectively. Data format: one-hot vector, dimension $[47, 53)$.

(c8). vipLevelName: Represents the VIP level of the user in Taobao. The meanings of values: 0 for unknown or no VIP level, 1~8 for VIP levels 1~8 respectively. Data format: one-hot vector, dimension $[53, 62)$.

(c9). phonePriceLevelPrefer: Represents the preferred phone price interval of the user. The meanings of values: 0 for unknown, 1 for 0~650 CNY, 2 for 1100~1700 CNY, 3 for 1700~1900 CNY, 4 for 1900~2600 CNY, 5 for 2600~3500 CNY, 6 for 3500~5000 CNY, 7 for 5000~8000 CNY, 8 for 650~1100 CNY, 9 for higher than 8000 CNY. Data format: one-hot vector, dimension $[62, 72)$.

(c10). zipCode: Represents the zip code of the address where the user is living. The meanings of values: The zip code contains 6 digits and each digit is a number from 0~9. We encode each digit with an one-hot vector and concatenate them together. Data format: one-hot vector, dimension $[72, 132)$.

(c11). idBirthyear: Represents the birthyear of the user. Data format: integer, dimension $[132, 133)$.

(c12). nationId: Represents the nation of the user. The meanings of values: 1 for China. (The real-world training data contains almost no data points from other countries. So does our generated data.) Data format: integer, dimension $[133, 134)$.

(c13). payOrdAmt: Represents the order amounts of the user in the last one month, one year, three months and six months. Data format: float numbers, dimension $[134, 138)$.

(c14). payOrdCnt: Represents the user's number of orders in the last one month, one year, three months and six months. Data format: integers, dimension $[138, 142)$.

(c15). payOrdDays: Represents the number of days when the user placed orders in the last one month, one year, three months and six months. Data format: integers, dimension $[142, 146)$.

(c16). payOrdItmCnt: Represents the number of item types the user bought in the last one month, one year, three months and six months. Data format: integers, dimension $[146, 150)$.

(c17). payOrdItmQty: Represents the number of items the user bought in the last one month, one year, three months and six months. Data format: integers, dimension $[150, 154]$.

(c18). pvAndIpv: Represents the PV and IPV value of the user bought in the last month. Data format: float numbers, dimension $[154, 156]$.

(c19). vstSlrCnt: Represents the number of sellers the user visited in the last month. Data format: integers, dimension $[156, 157]$.

(c20). vstCateCnt: Represents the number of categories the user visited in the last month. Data format: integers, dimension $[157, 158]$.

(c21). vstItmCnt: Represents the number of items the user visited in the last month. Data format: integers, dimension $[158, 159]$.

(c22). vstItmCnt: Represents the number of items the user visited in the last month. Data format: integers, dimension $[158, 159]$.

(c23). vstDays: Represents the number of days the user visited items in the last month. Data format: integers, dimension $[159, 160]$.

(c24). stayTimeLen: Represents the number of seconds the user spent on visiting items in the last month. Data format: integers, dimension $[160, 161]$.

(c25). cartItmCnt: Represents the number of items the user added to the cart in the last one week, two weeks, one month, three months, and six months. Data format: integers, dimension $[161, 166]$.

(c26). cltSlrCnt: Represents the number of sellers the user collected in the last one week, two weeks, one month, six months, and one year. Data format: integers, dimension $\{166, 168, 170, 172, 174\}$.

(c27). cltItmCnt: Represents the number of items the user collected in the last one week, two weeks, one month, six months, and one year. Data format: integers, dimension $\{167, 169, 171, 173, 175\}$.

**Structure of the value vector.** The value vector has 60 dimensions corresponding to 59 categories involved in our environment and one conserved category for the undefined or unknown category. The corresponding relations between the dimension indexes and categories are listed in Table 4.

## E   Tasks

Though AuctionNet provides a general framework for auto-bidding problem studies, we choose two typical scenarios in auto-bidding as tasks in AuctionNet for easier understanding.

### E.1   Basic Task

Our basic task is based on the scenario Budget Constrained Bidding (BCB) [33], where agents maximize their obtained values within the constraint on the budget. The optimization formulation of BCB from agent $i$'s perspective is as follows:

$$
\begin{aligned}
&\underset{\{\alpha_i^t\}}{\text{maximize}} \sum_{t=1}^{T} \left\langle \boldsymbol{x}_i^t, \boldsymbol{v}_i^t \right\rangle \\
&\text{s. t.} \sum_{t=1}^{T} \left\langle \boldsymbol{x}_i^t, \boldsymbol{c}_i^t \right\rangle \leq \omega_i,
\end{aligned}
\tag{2}
$$

where $\boldsymbol{x}_i^t = (x_{i1}^t, x_{i2}^t, \cdots, x_{im}^t)$ is the auction result of all ad opportunities for agent $i$ in time step $t$, $\boldsymbol{v}_i^t = (v_{i1}^t, v_{i2}^t, \cdots, v_{im}^t)$ is the value for agent $i$, $\boldsymbol{c}^t = (c_{i1}^t, c_{i2}^t, \cdots, c_{im}^t)$ is the cost in time step $t$, $b_i$ is the budget for agent $i$, and $\langle \cdot \rangle$ is the inner product.

As for the implementation, we know from our problem formulation that $r_i(s_t, \boldsymbol{a}_t) = \langle \boldsymbol{x}_i^t, \boldsymbol{v}_i^t \rangle$, so the objective in the optimization formulation is the same as the objective $\sum_{t=1}^{T} r_i(s_t, \boldsymbol{a}_t)$ in the RL formulation. The budget constraint is guaranteed by ignoring the bids exceeding agents' budgets in the environment. Therefore, BCB corresponds to the default setting of AuctionNet.

Table 4: Corresponding relations for categories.

| ID | Category | ID | Category |
|---|---|---|---|
| 1 | Snacks | 31 | Travel Services |
| 2 | Personal Care | 32 | Tmall Home & Living |
| 3 | Electric Vehicles | 33 | Maternity & Childcare |
| 4 | Tmall Underwear | 34 | Movies, Shows & Sports |
| 5 | Smart Toys & Games | 35 | Education & Teaching |
| 6 | Tea | 36 | Taobao Bags & Accessories |
| 7 | Household Cleaning | 37 | Taobao Underwear |
| 8 | Chilled Food | 38 | Audio & Video Electronics |
| 9 | Tmall Women's Clothing | 39 | Gaming |
| 10 | Enterprise Services | 40 | Pets |
| 11 | Dairy Products | 41 | Vehicles |
| 12 | Fragrances and Aromatherapy | 42 | Major Appliances |
| 13 | Life Services | 43 | Tmall Footwear |
| 14 | Household Appliances | 44 | Food Coupons |
| 15 | Taobao Men's Clothing | 45 | Auto Accessories |
| 16 | Tmall Home Decor | 46 | Mobile Phones |
| 17 | Taobao Home & Living | 47 | Taobao Footwear |
| 18 | Taobao Home Decor | 48 | Grains & Instant Food |
| 19 | Instant Drinks | 49 | Tmall Bags & Accessories |
| 20 | Alcohol | 50 | Mobile & Digital Accessories |
| 21 | Taobao Women's Clothing | 51 | Taobao Watches & Glasses |
| 22 | Auto Aftermarket | 52 | Jewelry & Accessories |
| 23 | Fruits and Vegetables | 53 | Sports |
| 24 | Flowers and Gardening | 54 | Toys & Fun |
| 25 | Office & School Supplies | 55 | Entertainment Recharge |
| 26 | Computers | 56 | Beverages |
| 27 | Tmall Watches & Glasses | 57 | Outdoor |
| 28 | Computer Accessories | 58 | Motorcycles |
| 29 | Aquatic Products, Meat, Poultry & Eggs | 59 | Tmall Men's Clothing |
| 30 | Cosmetics | 0 | Other |

## E.2  Target CPA Task

We propose Target CPA Task based on the real-world scenario Target CPA (Cost Per Action)[5] with some simplifications for understanding. The CPA of agent $i$ is defined as $\mathrm{cpa}_i = \frac{\sum_{t=1}^{T}\langle \boldsymbol{x}_i^t, \boldsymbol{c}_i^t \rangle}{\sum_{t=1}^{T}\langle \boldsymbol{x}_i^t, \boldsymbol{v}_i^t \rangle}$, which can be seen as the cost taken by agent $i$ for unit value. A low CPA means the budgets are consumed to obtain values effectively. Based on the basic task, Target CPA Task adds one more constraint on CPA that $\mathrm{cpa}_i$ should be lower than the desired CPA $d_i$. The formulation is as follows:

$$
\begin{aligned}
&\underset{\{\alpha_i^t\}}{\text{maximize}} \sum_{t=1}^{T} \left\langle \boldsymbol{x}_i^t, \boldsymbol{v}_i^t \right\rangle \\
&\text{s. t.} \sum_{t=1}^{T} \left\langle \boldsymbol{x}_i^t, \boldsymbol{c}_i^t \right\rangle \leq \omega_i \\
&\qquad \mathrm{cpa}_i \leq d_i.
\end{aligned}
\tag{3}
$$

Given that CPA can only be calculated at the end of one episode, the environment will only provide a sparse reward in Target CPA Task, which is different from the basic task. Unlike the budget constraint which cannot be violated in the environment, we allow agents to violate the CPA constraint, but we will penalize those agents for violations on their obtained values based on their CPA $\mathrm{cpa}_i$. The sparse reward formulation in Target CPA Task is as follows:

$$
r_i^{\mathrm{CSB}} = p(\mathrm{cpa}_i; d_i) \sum_{t=1}^{T} \left\langle \boldsymbol{x}_i^t, \boldsymbol{v}_i^t \right\rangle,
\tag{4}
$$

---

[5]https://support.google.com/google-ads/answer/6268632

where $p(\text{cpa}_i; d_i) = \min\left\{ \left(\frac{d_i}{\text{cpa}_i}\right)^\beta, 1 \right\}$ is the penalty function for exceeding the CPA constraint. The formulation of $p(\text{cpa}_i; d_i)$ implies that the penalty is incurred only when $\text{cpa}_i > d_i$. The parameter $\beta > 0$ is typically set to 3. Therefore, Target CPA Task can be implemented with modifications to the reward function in the basic task.

# F    Baseline Algorithms

We have implemented multiple baseline algorithms in AuctionNet to facilitate a quick start-up and comprehensive understanding of users. The baseline algorithms include PID Controller[36], Online LP[15], IQL[21], Behavior Cloning[30], and Decision Transformer[8].

**PID Controller.** PID Controller is a traditional algorithm in the control field with a long history[4]. It is simple but effective in many scenarios. Recently, PID Controller has also been adopted in online advertising[36]. The idea of PID Controller is straightforward: PID Controller takes three parameters $\lambda_P$, $\lambda_I$, and $\lambda_D$ for Proportional Control, Integral Control, and Derivative Control, respectively. We use the PID Controller to control the cost or bids of agents in this baseline.

**Online LP.** The optimization formulation (2) is a typical Linear Programming (LP) problem. Moreover, the variable $x_{ij}^t \in \{0, 1\}$ is binary, so the problem in each time step can be converted to a dynamic knapsack problem. Online LP solves this dynamic knapsack problem using a greedy algorithm.

**IQL.** Implicit Q-learning (IQL) is an offline RL algorithm. The idea of IQL is evaluating offline Q-function only on the actions that appeared in the offline data, to avoid the overestimation in the out-of-distribution data. In practice, IQL utilizes expectile regression to realize the offline Q-learning on in-distribution data.

**Behavior Cloning.** Behavior Cloning (BC) is a supervised learning algorithm given expert trajectories. The agent's policy learns by predicting the expert's action in the state of given trajectories. BC is a baseline for verifying the effectiveness of RL algorithms.

**Decision Transformer.** Decision Transformer (DT) utilizes the ability of Transformer[31] for sequential decision-making. DT views the trajectories in MDP as a sequence and predicts actions given previous transitions.

# G    Implementation and Modules

The environment of AuctionNet consists of three main modules: the ad opportunity generation module, the auction module, and the bidding module. The general process of one time step in AuctionNet can be concluded as follows:

1) The ad opportunity generation module generates features $\boldsymbol{u} = (u_1, u_2, \cdots, u_m)$ and values $\boldsymbol{v} = \{v_{ij}\}$ of $m$ ad opportunities for $n$ agents, where the number of ad opportunities $m$ is sampled from an intern distribution within AuctionNet. This intern distribution is obtained from real-world online advertising statistics

2) Agents bid for all the ad opportunities considering the predicted values provided by the environment and the historical auction logs.

3) The auction module determines the winner of each auction, rewards, and costs by the auction mechanism.

4) Agents receive rewards, costs, and new auction logs. The budgets of all the agents are updated according to auction results. In the next time step, all the processes above will be repeated.

Given this general process, we will introduce the three main modules in order. The ad opportunity generation module will generate features $\boldsymbol{u}$ of ad impressions related to the real online data. The ad auction module supports an auction similar to real-world online advertising and realizes several popular auction mechanisms for different research purposes. The bidding module supports explicitly modeling a multi-agent environment with several implemented baselines.

## G.1 Ad Opportunity Generation Module

The target of the ad opportunity generation module is to generate diverse ad opportunity features similar to real online advertising data. The core of this module is the generative model. The objective of the generative model in AuctionNet is to generate data resembling real advertising delivery data. Useful information in the real advertising delivery data can be divided into four parts: features of ad opportunities (users' information), features of advertisers, time when the ad opportunity arises, and the values of the ad opportunities. In our model, we simplify the feature of advertisers to be the advertisers' industry categories. We focus on the generation of ad opportunity features and take the categories and time as conditions. The generative model consists of two components: the generative model for ad opportunity features and the prediction model for the values.

**Feature Generation.** The ad opportunity feature contains two parts of information: the basic identity information of users and the consumption records such as the consumption amount. The identity information is discrete and the consumption records are continuous in general, which are processed with different measures. Diffusion [17] model is the most popular generative model recently which obtains SOTA performances in image generation with a simplified training process. We would like to adopt the diffusion model to generate the ad opportunity feature but struggle with the denoising operation which can result in unreasonable outputs such as a negative consumption amount. So we follow the idea of the Latent Diffusion Model (LDM)[25] to generate ad opportunity features. LDM adds noises and denoises in the latent space with a diffusion model and generates data from the latent space with an encoder and decoder. More details can be found in Appendix H.1.

**Value Prediction.** The value prediction model needs to handle three types of information: ad opportunity features, the industry category information of advertisers, and time information. We simplify the category and time information as discrete values. Therefore, we aim to integrate the category and time information into the ad opportunity features for better value prediction. Besides, we hope this integration can partly reflect the variation pattern of the impression values related to advertisers' features and time. Multi-head attention (MHA), as a popular network architecture and the critical part of Transformer [31], can capture the relations among a sequence, thus we hope to utilize MHA for better integration. We combine cross-attention and self-attention to integrate the three types of information. We also follow the idea of position embedding in the Transformer to process the time information. More details are included in Appendix H.2.

For the consideration of interaction efficiency in AuctionNet, the environment utilizes a dataset consisting of generated features and corresponding predicted values. More details of the dataset will be discussed in Section 4.1. Though the ad opportunity generation module is trained with real online advertising data, an important question is whether the generated data can reflect the properties of real data. Therefore, we have done several related experiments and the empirical results will also be discussed in Section 4.1.

## G.2 Auction Module

The task of the auction module is to determine the winner and the winning price given all bids of agents for the ad ad opportunities. The costs of agents will change given different auction rules. The most commonly discussed auction rule is the Generalized Second-Price (GSP) Auction which means the winner should pay a cost slightly higher than the second-highest bid instead of the highest bid. The auction module internally supports several popular auction rules including GSP for the convenience of researchers. Besides, researchers can also design a specific auction rule related to their purposes with the interface of the auction module.

Additionally, the property of multiple slots has been implemented in our simulation platform. Multiple slots emerge from the application in the industry, which means one ad opportunity has multiple ad slots for ad displays. The ad slots are ranked by their exposure rates. A higher exposure rate slot is more valuable for advertisers. Suppose the number of slots is $l$, then the auction module will attribute $l$ slots to the top $l$ bidders and these bidders will receive different values according to different exposure rates of slots. In the environment, $l$ is set to 3. Let $\text{slot}_{ij}$ represent the slot of ad opportunity $j$ wined by agent $i$ and $e_{ij} \in [0, 1]$ represent the exposure rate of $\text{slot}_{ij}$, then the optimization formulation of BCB with multiple slots is as follows:

$$\underset{\{\alpha_i^t\}}{\text{maximize}} \sum_{t=1}^{T} \sum_{j=1}^{m} e_{ij}^t x_{ij}^t v_{ij}^t$$

$$\text{s. t.} \sum_{t=1}^{T} \sum_{j=1}^{m} e_{ij}^t x_{ij}^t c_{ij}^t \leq \omega_i, \tag{5}$$

In summary, the multiple slots property increases the complexity of the optimal bidding strategy, since the exposure rate is a discount factor for both the cost and values. For instance, a strategy using a lower budget to bid for slots with relatively lower ranks may be better than the strategy that always chases the highest value slot. We believe supporting multiple slots in AuctionNet will be beneficial to reducing the gap between related research and the real-world online advertising platforms.

### G.3 Bidding Module

The bidding module is responsible for processing the multi-agent interactions between advertisers. This module implements the budget constraint and models the auto-bidding problem with sequence decision-making. Therefore, AuctionNet supports the mainstream paradigms including Budget Constrained Bidding (BCB) [33] and Multiple Constraints Bidding (MCB) [16] in the auto-bidding field. This will help researchers validate and gain insights from existing algorithms.

In the bidding module, we explicitly model the multi-agent setting. Researchers can implement multi-agent algorithms to achieve competition or cooperation among different agents. The varying bidding strategies of other agents can better reflect the complex and dynamic auction environment in real-world online advertising platforms. Besides, researchers can only control a part of the agents in AuctionNet while others is uncontrollable. This scenario is closer to the real advertising platform. The multi-agent setting of AuctionNet can adapt to different research objectives.

There are several different metrics for different business goals of advertisers in online advertising platforms such as Return-on-Investment (ROI) and Return-On-Ad-Spend (ROAS). AuctionNet has several built-in metrics covering the popular metrics used by the major advertising platform. Researchers can adopt these metrics conveniently to evaluate the performances of their auto-bidding strategies. Besides, researchers can define customized metrics according to their research objectives. Additionally, several popular algorithms in the auto-bidding field have been implemented as baselines in AuctionNet, including PID Controller[36], Online LP[15], IQL[21], Behavior Cloning[30], and Decision Transformer[8]. This can facilitate the interested researchers to quickly start up and evaluate these baselines in a unified environment.

## H Details of Deep Generative Networks

### H.1 Ad Opportunity Generation

The ad opportunity feature contains two parts of information: one is the basic identity information of users including gender, age, address and so on; another is the consumption records such as the consumption amount and the number of orders in the last three months. The identity information consists of discrete fields and each field has several candidates. The consumption records are continuous in general. Therefore, we process these two types of information with different measures.

Diffusion [17] model is the most popular generative model recently which obtains SOTA performances in image generation with a simplified training process. The principle of the diffusion model is adding Gaussian noises to original data in training and denoising from Gaussian noises in the generation process. We would like to adopt the diffusion model to generate the ad opportunity feature but struggle with the denoising operation which can result in unreasonable outputs such as a negative consumption amount. So we follow the idea of the Latent Diffusion Model (LDM)[25]. LDM has a latent space to encode the original data. LDM combines the idea of diffusion model with VAE[20]. LDM adds noises and denoises in the latent space with a diffusion model and generates data from the latent space with an encoder and decoder.

Specifically, let $U \subset \mathbb{R}^d$ be the space of ad opportunity feature data $(u_1, u_2, \cdots, u_K)$ where $d$ is the dimension of original data and $K$ is the volume of the ad opportunity. Let $Y \subset \mathbb{R}^{d'}$ be the latent

space $(d > d')$. The encoder and decoder are represented as $g_\phi$ and $h_\psi$ respectively, where $\phi$ and $\psi$ are the parameters. The function of the encoder $g_\phi$ is obtaining a latent representation of original data as follows:

$$g_\phi(u_k) = (\mu_k, \sigma_k), \quad y_k \sim \mathcal{N}(\mu_k, \sigma_k^2),$$

where $y_k \in Y$ is the latent representation. In practice, the reparameterize trick [20] is applied to make sure this operation is differentiable in the backpropagation. Given the latent representation $y_k$, the decoder is responsible for reconstructing the original data from $y_k$, *i.e.*, $h_\psi(y_k) = \tilde{u}_k \in U$. Besides the reconstruction, the latent distribution $\mathcal{N}(\mu_k, \sigma_k^2)$ is expected to be close to the standard Gaussian distribution $\mathcal{N}(0, 1)$. Therefore, we have the following loss function for the encoder and decoder:

$$\mathcal{L}_{recons} = \frac{1}{K} \sum_{k=1}^{K} \|u_k - h_\psi(y_k)\|_2^2, \quad \mathcal{L}_{reg} = \frac{1}{K} \sum_{k=1}^{K} D_{\mathrm{KL}}\left(\mathcal{N}(\mu_k, \sigma_k^2)\|\mathcal{N}(0, 1)\right),$$

where $\mathcal{L}_{recons}$ is the reconstruction loss and $\mathcal{L}_{reg}$ is the regularization loss for the latent distribution.

Different from the original idea of VAE, where the latent variable $y \in Y$ is sampled from $\mathcal{N}(0, 1)$ in the generation process, LDM uses a diffusion model in the latent space to generate the latent variable. In general, the idea of the diffusion model is adding Gaussian noises to the original data to obtain variables in $\mathcal{N}(0, 1)$ and denoising from $\mathcal{N}(0, 1)$ for generation. Given a latent variable $y$, we denote the noisy version of $y$ after $p$ iterations as $y_p$. The diffusion model has a network to predict noise $\epsilon_\theta(y_p, p)$ and the loss function can be represented as

$$\mathcal{L}_{LDM} = \frac{1}{K} \sum_{k=1}^{K} \|\epsilon - \epsilon_\theta(y_{k,p_k}, p_k)\|_2^2,$$

where $\epsilon \sim \mathcal{N}(0, 1)$, $y_k$ is the latent embedding of $u_k$ and $p_k$ is uniformly sampled from the set $\{1, 2, \cdots, p_{\max}\}$. $\epsilon_\theta(y_p, p)$ is the only learnable network in the diffusion model, with which the process of adding noises and denoising can be completed by the basic operations.

As for the generation process, a latent variable $\bar{y}$ is sampled from $\mathcal{N}(0, 1)$ and $\tilde{y}$ is obtained by $t_{\max}$ denoising steps from $\tilde{y}$ given the noise prediction network $\epsilon_\theta$. Finally, the decoder generates an ad opportunity feature based on $\tilde{y}$ as $\tilde{x} = h_\psi(\tilde{y})$.

### H.2  Value Prediction

The value prediction model needs to handle three types of information: ad opportunity features, category information and time information. The category information corresponds to the industry categories of advertisers and the time information corresponds to the time when the ad opportunity arrived. In our model, the category and time information are simplified as discrete values. Therefore, we aim to integrate the category and time information into the ad opportunity features for better value prediction. Besides, we hope this integration can partly reflect the variation pattern of the impression values related to advertisers' features and time.

Multi-head attention (MHA) [31] is a popular network architecture and the critical part of Transformer [31]. MHA can capture the relations among a sequence, thus we hope to utilize MHA for better integration. The formulation of the attention network is straightforward as $\mathrm{Attention}(Q, K, V) = \mathrm{softmax}(\frac{QK^T}{\sqrt{d}}) \cdot V$. Multi-head attention can be viewed as applying the attention network in different representation subspaces as follows:

$$\mathrm{MultiHead}(Q, K, V) = \mathrm{Concat}(\mathrm{head}_1, \mathrm{head}_2, \cdots, \mathrm{head}_h)W^O, \tag{6}$$

$$\text{where } \mathrm{head}_i = \mathrm{Attention}(QW_i^Q, KW_i^K, VW_i^V). \tag{7}$$

$W_i^Q, W_i^K, W_i^V$ are the parameters for the projection networks of head $i$ and $W^O$ is the output network parameters of the MHA model.

We combine cross-attention and self-attention to integrate the three types of information. Suppose $u_k$, $u_k^{\mathrm{time}}$ and $q_k$ are the ad opportunity feature, time information and category information in a single

record respectively, then we will process the information as follows:

$$Q^{(1)} = \tau_Q^{(1)}(u_k^{\text{time}}), \quad K^{(1)} = \tau_K^{(1)}(u_k), \quad V^{(1)} = \tau_V^{(1)}(u_k),$$

$$z_k^{(1)} = \text{MultiHead}(Q^{(1)}, K^{(1)}, V^{(1)}),$$

$$Q^{(2)} = \tau_Q^{(2)}(z_k^{(1)}), \quad K^{(2)} = \tau_K^{(2)}(z_k^{(1)}), \quad V^{(2)} = \tau_V^{(2)}(z_k^{(1)}),$$

$$z_k^{(2)} = \text{MultiHead}(Q^{(2)}, K^{(2)}, V^{(2)}),$$

$$Q^{(3)} = \tau_Q^{(3)}(q_k), \quad K^{(3)} = \tau_K^{(3)}(z_k^{(2)}), \quad V^{(3)} = \tau_V^{(3)}(z_k^{(2)}),$$

$$z_k^{(3)} = \text{MultiHead}(Q^{(3)}, K^{(3)}, V^{(3)}),$$

$$Q^{(4)} = \tau_Q^{(4)}(z_k^{(3)}), \quad K^{(4)} = \tau_K^{(4)}(z_k^{(3)}), \quad V^{(4)} = \tau_V^{(4)}(z_k^{(3)}),$$

$$z_k = \text{MultiHead}(Q^{(4)}, K^{(4)}, V^{(4)}),$$

where $\tau^{(1)}, \tau^{(2)}, \tau^{(3)}, \tau^{(4)}$ are the projection function for the multi-head attention network.

The variation of ad opportunity values has some temporal patterns in the real world. Therefore, we follow the position encoding idea in Transformer [31] and Diffusion Model [17] to process the time information. Let $\text{PE} : \mathbb{N} \to \mathbb{R}^d$ represent the position encoding function, then

$$\text{PE}_{2s}(t) = \sin\left(\frac{t}{10000^{\frac{2s}{d}}}\right), \quad \text{PE}_{2s+1}(t) = \cos\left(\frac{t}{10000^{\frac{2s}{d}}}\right),$$

where $t$ is the discrete time and $s$ corresponds to the dimension in the embedding $\text{PE}(t)$. Let $e_k = \text{PE}(u_k^{time})$, then the value prediction is conducted by $\hat{v}_k = U_\xi(z_k, e_k)$, where $U_\xi(z_k, e_k)$ is the prediction network with a similar architecture to the U-Net [26] used by the Diffusion Model [17]. The loss of the value prediction model is shown below:

$$\mathcal{L}_{\text{pred}} = \frac{1}{N} \sum_{k=1}^{N} \|v_k - \hat{v}_k\|_2^2,$$

where $v_k$ is the true value of the ad opportunity in the record of $u_k$.

