# OpenReview forum: "AuctionNet: A Novel Benchmark for Decision-Making in Large-Scale Games"
_NeurIPS.cc/2024/Datasets_and_Benchmarks_Track — NeurIPS 2024 Track Datasets and Benchmarks Spotlight_

### Official Review · Reviewer_UBgR · 2024-06-26
**This paper provides the LSA Environment, a realistic data-driven multi-agent simulation environment for auto-bidding augmented with a deep generative model.**

**Rating:** 6
**Confidence:** 5
**Correctness:** Yep
**Clarity:** good, can be polished

**Review:**

Strongness
[S1] The motivation of this paper is clear, and auto-bidding in advertising auctions is indeed a very popular and important topic nowadays.
[S2] Diverse auto-bidding algorithms are evaluated using the proposed benchmark which not only justifies the effectiveness of the work but also offers some significant insights for future research.
[S3] This work is complete, including the auto-bidding algorithm, LSA Environment augmented with a deep generative model, a very large dataset, and relatively sufficient experiments.
[S4] Diverse auto-bidding algorithms are evaluated using the proposed benchmark which not only justifies the effectiveness of the work but also offers some significant insights for future research.

Weakness
[W1] The writing of some parts of the article can be improved, and specific issues are provided in the following details [D1] and [D2].
[W2] For the part of your problem formulation, In lines 82-84, I am very interested in why the \alpha_i is a linear coefficient, is there a possibility of other forms? For the transition function, is this a degenerate state transition function, as it is deterministic and lacks randomness?
[W3] You use a deep generative mode to generate data in the environment and compare the generated data with real data. I am very interested in why not instead directly use real feature data in the LSA environment.
[W4] For the evaluations of algorithms, why didn’t you simulate the AdCraft and AuctionGym which you mentioned multiple times in this article?

**Strengths:**

pls see Review

**Additional Feedback:**

NA

**Documentation:**

yep

**Limitations:**

Yep

**Opportunities For Improvement:**

[D1] The process shown in Figure 1 should provide some explanations, especially for some proprietary terms such as CPA, conversion, etc., which are explained later in the text, but I believe that explanations should be provided here so that readers can understand.
[D2] In lines 93-94, it will be better to explain the specific meaning of “partial information” in your model, because information is a a key element in game theory, and knowing what information is incomplete is important for understanding your model, such as \alpha_i, v_i of other participants.

**Relation To Prior Work:**

yep

**Summary And Contributions:**

This paper provides the LSA Environment, a realistic data-driven multi-agent simulation environment for auto-bidding augmented with a deep generative model. They provide the LSA Dataset, which comprises over 500 million records, totaling 40 GB in size, and contains trajectories with 50 diverse agents competing with each other for effective offline training. Finally, they evaluate different types of algorithms in LSA and verify the similarity between the generated data and the real-world data.

---

> ### Author Rebuttal · Authors · 2024-08-16
>
> Thank you for your advice and comments.
>
> > [W1] The writing of some parts of the article can be improved, and specific issues are provided in the following details [D1] and [D2].
> >
> > [D1] The process shown in Figure 1 should provide some explanations, especially for some proprietary terms such as CPA, conversion, etc., which are explained later in the text, but I believe that explanations should be provided here so that readers can understand.
> >
> > [D2] In lines 93-94, it will be better to explain the specific meaning of “partial information” in your model, because information is a a key element in game theory, and knowing what information is incomplete is important for understanding your model, such as \alpha_i, v_i of other participants.
>
> We will provide more explanations in the caption of Figure 1 in the revision for [D1]. As for [D2], specifically, the observation of agent $i$ $o_i = (b_i, \boldsymbol{f}, \boldsymbol{v})$. As for actions, Agents can only observe its own action $\alpha_i$ and the bidding results for each impression $x_i^j$. We will add these explanations in the revision.
>
> > [W2] For the part of your problem formulation, In lines 82-84, I am very interested in why the \alpha_i is a linear coefficient, is there a possibility of other forms? For the transition function, is this a degenerate state transition function, as it is deterministic and lacks randomness?
>
> From the perspective of an individual agent, the bidding problem can be formulated as follows:
>
> $\max_{x_i} \sum_i x_i v_i$
>
> $s.t. \sum_i x_i c_i \le B$
>
> $\quad \quad 0 \le x_i \le 1$
>
> where index $i$ represents $i$-th impression opportunity. By the primal-dual method, the optimal solution has the formulation $bid_i^* = \alpha^* v_i$, where $\alpha^*$ is the optimal bid rate.
>
> Therefore, taking a bid rate $\alpha_i$ as the action of agent $i$ is a common practice for researches in auto-bidding and more detailed discussions can be found in [1,2,3].
>
> As for the issue of randomness, the transition function of the LSA environment is not degenerate since the volume and values of the impression opportunities have randomness in each step. Additionally, the bidding strategies of other agents also have a degree of randomness which will also influence the state transition. We will add these explanations in the revision.
>
> [1] Xun Yang, Yasong Li, Hao Wang, Di Wu, Qing Tan, Jian Xu, and Kun Gai. 2019. Bid optimization by multivariable control in display advertising. In Proceedings of the 25th ACM SIGKDD International Conference on Knowledge Discovery & Data Mining. ACM, 1966–1974.
>
> [2] Weinan Zhang, Shuai Yuan, and Jun Wang. 2014. Optimal real-time bidding for display advertising. In Proceedings of the 20th ACM SIGKDD International Conference on Knowledge Discovery and Data Mining. ACM, 1077–1086.
>
> [3] Zhang, Haoqi, et al. "A personalized automated bidding framework for fairness-aware online advertising." *Proceedings of the 29th ACM SIGKDD Conference on Knowledge Discovery and Data Mining*. 2023.
>
> > [W3] You use a deep generative mode to generate data in the environment and compare the generated data with real data. I am very interested in why not instead directly use real feature data in the LSA environment.
>
> This is a critical question. We propose the LSA environment with the objective of facilitating related researchers with a simulation platform based on a realistic problem. However, the real feature data in online advertising involves with issues of sensitive information, personal privacy and security. We also hope the LSA environment can reflect some properties of the real world auto-bidding problem. Therefore, we choose to use deep generative model to synthesize similar data for the LSA environment. Moreover, the scale of the real data is enormous, potentially reaching hundreds of millions of records. Using a generative model is more beneficial for the platform's lightweight design compared to directly using real data.
>
> > [W4] For the evaluations of algorithms, why didn’t you simulate the AdCraft and AuctionGym which you mentioned multiple times in this article?
>
> In our evaluations, the agent will compete against competitors pre-trained by different algorithms in several episodes. In each episodes, the agent will represent a different advertiser with a different budget. AuctionGym can't support this evaluation method for the reason that it doesn't consider the budget. AdCraft can't support this evaluation method for the reason that its competitors' policies can't be customized as agents from different algorithms. We think our evaluation method can present the ability of different algorithms more comprehensively and is more close to the real-world auto-bidding problem. Therefore, it may not be necessary to simulate AdCraft and AuctionGym for evaluation in our opinion.

---

> > ### Comment · Reviewer_FJBQ · 2024-08-27
> >
> > Acknowledged. Thanks!

---

### Official Review · Reviewer_qWub · 2024-07-12
**The created dataset is sizeable, but the clarity of the presentation and strength of the exposition can be improved.**

**Rating:** 6
**Confidence:** 3

**Review:**

While the motivation does make sense, and the benchmark (specificall its two components, the simulation environment and the dataset) do appear to be sound. The presentation and exposition can both be improved, detailed in the following parts "opportunities for improvement" and "clarity", respectively.

**Strengths:**

- The considered benchmark does have real-world motivations.
- The collected dataset is of a significant size.

**Additional Feedback:**

The authors mention that the dataset and the environment are involved in a large compeition to be held in NeurIPS 2024, so they cannot disclose it now, which is understanable. However, this makes it more challenging to verify the detailed documentation such as by skimming through the dataset.

**Clarity:**

Regarding the presentation, some parts of writing/presentation can be improved.

In lines 243-244,
> Furthermore, the reward drops of all baselines, especially BC and Abid, in the CSB tasks are caused by the CPA penalty in (3) for exceeding the constraint.

It is not so clear why this is the case.

The fonts in Figures are too small.

Figures 6 & 7 appear before Figure 5 in main text.

In lines 173,
> Figure 3 that the impression values of different categories exhibit distinct patterns of variation.

How can one interpret such patterns of variation?

In lines 255-256,
> these agents’ policies are sampled from some parameterized distributions, which cannot reflect the multi-agent property essence of this problem.

What is the rationale? and how does this work (benchmark) addresses that?


In lines 256-257,
> Despite the discussion above, these existing simulation environments lack a data-driven method for modeling the real online advertising system.

What is the distinct benefit of a data-driven method?

**Correctness:**

The evaluation of "to show that generated data resembles real-world data in general." can be improved

In lines 195-196,
> The 2D PCA results are illustrated in Figure 5 and show that most generated data points overlap with real-world data points.

It seems there are over 175 dimensions in the original data. Then, can a 2D PCA produce a reliable result for comparing the similarity between the real-world data and generated data? It seems a more sophisticated technique or a higher dimension of PCA may be better suited.


The evaluation in Section 4.2 can be improved.

In lines 228-229,
> The empirical results show that the variation trends over the changes in category and time of predictions are similar to the ground truth in general.

Is there a quantitative result? and is there a some form of comparison available to show that this prediction model is accurate?

**Documentation:**

There is description of the benchmark in the main paper, but there is no open-source or attached file for the documentation.

**Limitations:**

Limitations are discussed in Section 6.

**Opportunities For Improvement:**

Regarding the exposition, there seems to be room for better justification of some approaches.

In lines 38-39,
> which inspires us to model the distribution of real-world impressions with a deep generative model.

Is there additional justification?

In line 88,
> $r_i(s,\mathbf{a})= \sum_{j=1}^m x_i^j v_i^j$

The reward is solely determined by the values and whether the bid is successful. Why is it that the reward does not depend on the amount of the bid?

In line 89,
> cost $c_j$ is determined by the winning price

Is the cost c_j is exactly the winning bid? Is there a reason not to use second-price auction?

For Equation (3), is there a reference or justification for this precise formulation?

In lines 174,
> agents should consider the impression value variation over time to bid for appropriate impressions at the appropriate time.

Precisely how should the agents carry out such considerations?

**Relation To Prior Work:**

Some discussion with prior work is presented in Section 5 and it does appear that this work differs from those mentioned in Section 5.

**Summary And Contributions:**

The authors describe a large-scale auction (LSA) benchmark, consisting of a multi-agent simulation environment and a dataset. The environment is augmented by a deep generative model. The dataset consist of over 500 million records, and 50 trajectories of agents interacting/competing with each other. The authors evaluate different existing auot-bidding algorithms.

---

> ### Author Rebuttal · Authors · 2024-08-16
>
> Thank you for your comments.
>
> > In lines 38-39,
> >
> > > which inspires us to model the distribution of real-world impressions with a deep generative model.
> >
> > Is there additional justification?
>
> There are several existing works using deep generative model for synthetic data of real-word applications. [1] uses GAN for the generation of electronic health records in medical studies. [2] applies deep generative model for data generation of health applications sensory data. Deep generative model has also been applied for mobility datasets generation, which are are fundamental for evaluating algorithms pertaining to geographic information systems [3]. Notably, parts of their motivations are security and sensitive information which are similar to ours. We will update these discussions in the revision.
>
> [1] M. K. Baowaly, C.-C. Lin, C.-L. Liu, and K.-T. Chen, ‘‘Synthesizing electronic health records using improved generative adversarial networks,’’ J. Amer. Med. Inform. Assoc., vol. 26, no. 3, pp. 228–241, Mar. 2019.
>
> [2] S. Norgaard, R. Saeedi, K. Sasani, and A. H. Gebremedhin, ‘‘Synthetic sensor data generation for health applications: A supervised deep learning approach,’’ in Proc. 40th Annu. Int. Conf. IEEE Eng. Med. Biol. Soc. (EMBC), Jul. 2018, pp. 1164–1167.
>
> [3] V. Kulkarni, N. Tagasovska, T. Vatter, and B. Garbinato, ‘‘Generative models for simulating mobility trajectories,’’ CoRR, vol. abs/1811.12801, 2018. [Online]. Available: https://arxiv.org/abs/1811.12801
>
> > In line 88,
> >
> > > $r_i(s,\mathbf{a})= \sum_{j=1}^m x_i^j v_i^j$
> >
> > The reward is solely determined by the values and whether the bid is successful. Why is it that the reward does not depend on the amount of the bid?
>
> The value $v^j_i$ corresponds to the conversion probability of the impression opportunity $j$ for advertiser $i$ in the real world, *e.g.*, the probability of an user $j$ clicking on the ad provided by advertiser $i$. Therefore, the value doesn't depend on the amount of the bid, which will influence the cost $c_j$. Under the budget constraints, if the cost $c_j$ is too high, it will result in acquiring a smaller amount of impression opportunities, leading to a decrease in total value. Therefore, we think this reward can comprehensively consider both the value and cost factors in auto-bidding.
>
> > In line 89,
> >
> > > cost $c_j$ is determined by the winning price
> >
> > Is the cost c_j is exactly the winning bid? Is there a reason not to use second-price auction?
>
> In the second-price auction, $c_j$ is the second highest bid among all agents. In line 642~648, we have stated that the LSA environment support multiple auction rules including GSP. In general, GSP has many advantages and been applied widely in online advertising. However, there are still some reasons of not using second-price auction in realistic applications. One reason is that the first-price auctions are more transparent for advertisers, as the second-price auctions may cause the issue of hidden fees [1]. The platform may need to bear the additional explanation costs. Another reason may be that the second-price auction may cause income decreases of the advertising platform [2]. Additionally, GSP, as a fixed rule, can't dynamically balance the performance of users, advertisers and the platform based on changes in the environment. This can easily lead to imbalances, such as a decline in advertiser performance or a loss of platform revenue. Therefore, more advanced methods such as Neural Auction are proposed for improvement [3].
>
> [1] Weitong Ou, Bo Chen, Xinyi Dai, Weinan Zhang, Weiwen Liu, Ruiming Tang, and Yong Yu. 2023. A Survey on Bid Optimization in Real-Time Bidding Display Advertising. ACM Trans. Knowl. Discov. Data 18, 3, Article 58 (April 2024), 31 pages. [A Survey on Bid Optimization in Real-Time Bidding Display Advertising | ACM Transactions on Knowledge Discovery from Data](https://doi.org/10.1145/3628603) [2] Ross Benes. First-price auctions are driving up ad prices. [First-Price Auctions Are Driving Up Ad Prices](https://www.emarketer.com/content/first-price-auctions-are-driving-up-ad-prices), 2018. Published: October 17, 2018
> [3] Liu X, Yu C, Zhang Z, et al. Neural auction: End-to-end learning of auction mechanisms for e-commerce advertising[C]//Proceedings of the 27th ACM SIGKDD Conference on Knowledge Discovery & Data Mining. 2021: 3354-3364.
>
> > For Equation (3), is there a reference or justification for this precise formulation?
>
> Target CPA is popular in both real-world applications and researches of auto-bidding [1,2,3]. The CPA constraint will encourage cost-effective strategies. We think it is intuitive to change the hard CPA constraint into a soft penalty related to CPA in optimization for simplicity, as it will reduce the difficulty of training with a denser reward signal.
>
> [1] Junwei Lu, Chaoqi Yang, Xiaofeng Gao, Liubin Wang, Changcheng Li, and Guihai Chen. 2019. Reinforcement learning with sequential information clustering in real-time bidding. In Proceedings of the 28th ACM International Conference on Information and Knowledge Management. 1633–1641.
>
> [2] Nyoman Gunantara. 2018. A review of multi-objective optimization: Methods and its applications. Cogent Engineering 5, 1 (2018), 1502242.
>
> [3] Pingzhong Tang, Xun Wang, Zihe Wang, Yadong Xu, and Xiwang Yang. 2020. Optimized cost per mille in feeds advertising. In Proceedings of the 19th International Conference on Autonomous Agents and MultiAgent Systems. 1359– 1367.

---

> > ### Author Rebuttal · Authors · 2024-08-16
> >
> > > In lines 195-196,
> > >
> > > > The 2D PCA results are illustrated in Figure 5 and show that most generated data points overlap with real-world data points.
> > >
> > > It seems there are over 175 dimensions in the original data. Then, can a 2D PCA produce a reliable result for comparing the similarity between the real-world data and generated data? It seems a more sophisticated technique or a higher dimension of PCA may be better suited.
> >
> > We provide the 3D PCA result in https://sites.google.com/view/lsa-pca-3d . For better presentations, we use six different views in the 3D space. We can find that the generated data still overlaps with the original data in the 3D space. We will add these results in the revision.
> >
> > > > The empirical results show that the variation trends over the changes in category and time of predictions are similar to the ground truth in general.
> > >
> > > Is there a quantitative result? and is there a some form of comparison available to show that this prediction model is accurate?
> >
> > We calculate the earth mover's distances (EMD) between the distributions of the original and generated data over pCVR, pCTR and value in Figure 9. The EMDs are included in the following table and the distance between the original and generated distributions are relatively small. We think this can be a quantitative result to show the accuracy of the prediction model.
> >
> > |     | pCVR | pCTR | value |
> > | --- | --- | --- | --- |
> > | category | 0.00186 | 0.00225 | 0.000344 |
> > | time | 0.00118 | 0.000922 | 0.0000544 |
> >
> > Additionally, we need to argue that we have done some comparisons in Figure 9 and our approach of comparison in Figure 9 has been acknowleged by Reviewer 3s3b. The illustrations can show our model accurately predicts the trend of changes in pCTR, pCVR and value.
> >
> > To present the results more intuitively, we provide some extra quantitative results. We compare the MSE between the generated and original distribution with the standard deviation of original distribution. The quatitaive results are showed in the following table. It can be found that the MSEs are all smaller than the original_stds which means our prediction model can capture the patterns of the value variation and is accurate.
> >
> > |     | pCVR_category | pCTR_category | value_category | pCVR_time | pCTR_time | value_time |
> > | --- | --- | --- | --- | --- | --- | --- |
> > | MSE | **0.0341** | **0.0280** | **0.00496** | **0.0313** | **0.0259** | **0.00176** |
> > | original_std | 0.0685 | 0.0517 | 0.00573 | 0.0637 | 0.0590 | 0.00625 |
> >
> > > In lines 243-244,
> > >
> > > > Furthermore, the reward drops of all baselines, especially BC and Abid, in the CSB tasks are caused by the CPA penalty in (3) for exceeding the constraint.
> > >
> > > It is not so clear why this is the case.
> >
> > Here we add the statistics about the CPA penalty in the CSB tasks in the following table, where the CPA excess rate corresponds to $cpa_i - 1$. We can find that all the baselines violate the CPA constraints with different extents and are affected by the CPA penalty on their rewards. The reason for the severe affections on BC and Abid may be that these algorithms do not fully consider multi-constraint optimization, resulting in a high violation rate of the CPA constraint. We will update these results in the revision.
> >
> > | Metric | OnlineLP | IQL | DT  | PID | BC  | Abid |
> > | --- | --- | --- | --- | --- | --- | --- |
> > | mean original reward | 2182.081 | 1947.401 | 2140.933 | 2192.971 | 1097.351 | 1375.090 |
> > | mean penalized reward | 1958.545 | 1596.964 | 1896.789 | 1990.190 | 340.893 | 884.946 |
> > | mean CPA excess rate | 0.326 | 0.444 | 0.314 | 0.297 | 0.796 | 0.467 |
> >
> > > The fonts in Figures are too small.
> > >
> > > Figures 6 & 7 appear before Figure 5 in main text.
> >
> > Thank you for your advice. We will adjust the Figures in the revision.
> >
> > > In lines 173,
> > >
> > > > Figure 3 that the impression values of different categories exhibit distinct patterns of variation.
> > >
> > > How can one interpret such patterns of variation?
> >
> > We would like to show the variation of volumes and values for different categories in Figure 3. For instance, the impression opportunities of Category 2 may occur more frequently in both morning and afternoon, while the ones in the afternoon have higher values. Agents should capture and utilize these patterns to obtain a cost-effective bidding strategy.
> >
> > The phenomenon of different categories exhibiting distinct patterns is similar to the observations in the real-world auto-bidding problems. For instances, most impression opportunities of medical/office supplies occurs between 8 AM and 6 PM, as these are primarily company procurement activities. For women's fashion/beauty products, since young women typically have leisure time in the evenings and early mornings, the impression distribution also shows higher levels during these times.

---

> ### Author Rebuttal · Authors · 2024-08-16
>
> > In lines 255-256,
> >
> > > these agents’ policies are sampled from some parameterized distributions, which cannot reflect the multi-agent property essence of this problem.
> >
> > What is the rationale? and how does this work (benchmark) addresses that?
>
> We think the policies generated from some parameterized distributions are relative too simple to reflect properties of the auto-bidding problem in the real world. This approach also means the other agents' policies will not change in the bidding process, so they can be seen as a part of the environment. Moreover, such static policies in AdCraft can not reflect the challenges posed by the competitive interactions and dynamic adaptions of agents' policies in real world auto-bidding problems.
>
> Our LSA environment provides a more general and flexible interface like OpenAI Gym for the agents' policies, which supports different types of bidding strategies including rule-based policies, linear programming policies and neural network policies from different algorithms. Moreover, LSA environment allows the policy updates of multiple agents at the same time, which is more able to reflect the multi-agent property essence of the auto-bidding problem in our opinion. More details can be found in Appendix E.
>
>
> > In lines 256-257,
> >
> > > Despite the discussion above, these existing simulation environments lack a data-driven method for modeling the real online advertising system.
> >
> > What is the distinct benefit of a data-driven method?
>
> As we have stated in the introduction part, our objective is to provide a realistic large-scale game environment and we choose the auto-bidding problem as the representative. We think the real-world auto-bidding problem is complex and can not be described or modeled by some simple principles, to the best of our knowledge. Therefore, the data-driven method, especially supported by the real-world data or similar synthetic data, may be more convincing to reflect the properties of the real-world problem. The researches and studies done in such a environment may be more meaningful to the real-world applications in our opinion.
>
> > There is description of the benchmark in the main paper, but there is no open-source or attached file for the documentation.
>
> As for the open-source or attached file for the documentation, we plan to release these materials at a future date after we have refined the documentation.

---

> ### Author Response · Authors · 2024-08-22
> **Follow-up**
>
> Thank you for your thorough review and valuable feedback on our manuscript.  We think we have addressed your comments and provided additional empirical results.  As the review process is nearing its conclusion, we would greatly appreciate it if you could provide your further feedback at your earliest convenience.

---

> > ### Comment · Reviewer_qWub · 2024-08-28
> > **Acknowledgement**
> >
> > I would like to thank the authors for providing a detailed rebuttal. Most of my questions were answered and I have increased my rating.

---

### Official Review · Reviewer_3s3b · 2024-07-24

**Rating:** 7
**Confidence:** 3
**Clarity:** Yes, the paper is clearly written.

**Review:**

The proposed benchmark focuses on making the simulation more realistic compared to existing benchmarks. This is done by simulating multiple agents' bidding and introducing budget constraints, and the qualitative analysis shows that the proposed benchmark simulates real-world interactions well. Additionally, the publicized synthetic data is also large-scale, which is beneficial for real-world applications.

Potential improvement relies on the documentation, and providing detailed instructions for customizing the environment would be valuable for users (see opportunities for improvement in detail).

**Strengths:**

- The paper demonstrates that the statistics of the synthetic data correlate with those of real-world datasets, showing that the simulation is reasonably realistic.

- The problem formulation, including the settings of state and action spaces, seems reasonable. Also, the budget constraints and multiple agent simulation replicate practical situations well.

- Although simulating interactions with multiple agents, the user interface follows Gym format and can be easily used by users.

**Additional Feedback:**

NA.

**Correctness:**

- The way the paper shows that the synthetic data has correlated statistics with the real-world dataset seems beneficial. In particular, Figure 8 and 9 demonstrate that a variety of statistics, such as consumption, CTR, CVR are simulated well.

- The chosen algorithms and experiment design follow the standard.

**Documentation:**

- The example code and details of the dataset are provided in the paper (main text and appendix).

- It would be useful if there were documentation on (i) how to customize each module with varying configurations and (ii) how to use users' own real-world datasets to simulate the traffic and interactions.

**Ethics:**

NA.

**Limitations:**

- See an opportunity for improvement.

- A potential concern is that because the benchmark uses a simulated environment, it is not guaranteed that the simulation works well under the distribution shift.

**Opportunities For Improvement:**

Additional documentation on how to customize the environment and how to use the users' own data to define simulation would be beneficial for promoting real-world applications.

**Relation To Prior Work:**

The cited benchmarks are representative ones, and the paper claims that the proposed benchmark is novel in that it simulates multiple agents (bidders) and budget constraints.

**Summary And Contributions:**

This paper presents a new simulation-based benchmark for online advertisement (bidding). The proposed benchmark and standardized dataset is more large-scale and realistic compared to existing benchmarks in that it simulates (i) multiple bidding agents and (ii) budget constraints. Also, the qualitative comparison between the real and synthetic data suggests that the synthetic data generated by the proposed benchmark has statistics similar to those of real-world datasets.

---

> ### Author Rebuttal · Authors · 2024-08-16
>
> Thank you for your positive comments.
>
> > Additional documentation on how to customize the environment and how to use the users' own data to define simulation would be beneficial for promoting real-world applications.
>
> As for the documentation and tutorials, we plan to release these materials at a future date after we have refined the documentation.
>
> > A potential concern is that because the benchmark uses a simulated environment, it is not guaranteed that the simulation works well under the distribution shift.
>
> This is a critical question. Simulated environments has been widely applied in the field of RL and has also promoted the advancement in related researches [1,2,3,4]. Experiments of auto-bidding algorithms in the real online advertising environment are expensive and difficult, sometimes involved with the risk of sensitive data exposure. Therefore, the simulated environment can be a solution to avoiding the risks and cost in online experiments.
>
> We have recognized the issue of the gap between the simulated environment and real-world tasks. Therefore we try to minimize the gap with a deep generative model trained by real-world data. We show the similarity between the generated and real data by empirical results from multiple perspectives. This similarity, to some extents, allow us to believe that the impact of distribution shift is minimal.
>
> Though the gap issue may not be completely resolved in this paper, at least we take one step further in minimizing the gap and can provide some insights for related researches.
>
> [1] Samvelyan, Mikayel, et al. "The starcraft multi-agent challenge." *arXiv preprint arXiv:1902.04043* (2019).
>
> [2] Ellis, Benjamin, et al. "Smacv2: An improved benchmark for cooperative multi-agent reinforcement learning." *Advances in Neural Information Processing Systems* 36 (2024).
>
> [3] Emanuel Todorov, Tom Erez, and Yuval Tassa. Mujoco: A physics engine for model-based control. In IEEE/RSJ International Conference on Intelligent Robots and Systems (IROS), 2012.
>
> [4] Georgios Papoudakis, Filippos Christianos, Lukas Schäfer, and Stefano V Albrecht. Benchmarking multi-agent deep reinforcement learning algorithms in cooperative tasks. In Advances in Neural Information Processing Systems (NeurIPS), 2021.

---

> > ### Comment · Reviewer_3s3b · 2024-08-27
> >
> > Thank you for the responses. It is great to know the plans for the update of the proposed platform and the measurements for the potential distribution shifts. I look forward to the official publication of the benchmark.

---

### Official Review · Reviewer_FJBQ · 2024-07-26
**Large-Scale Auction (LSA) Benchmark -  A Simulation for Decision-Making in Competitive Online Advertising**

**Rating:** 8
**Confidence:** 4
**Correctness:** Yes
**Clarity:** Yes

**Review:**

The paper "A Novel Benchmark for Decision-Making in Uncertain and Competitive Games" presents a significant advancement in the study of decision-making in large-scale, competitive environments. By developing the LSA Environment and accompanying dataset, the authors address a critical gap in the availability of realistic, comprehensive game environments for research. The detailed evaluation of various baseline algorithms provides valuable insights into their performance and potential areas for improvement. The focus on real-world applicability, particularly in the context of online advertising, enhances the relevance and impact of this work. Overall, the LSA Benchmark is a robust tool that can drive future research and innovation in decision-making technologies, offering a more realistic and practical framework for studying complex, multi-agent interactions.

**Strengths:**

The strengths in the paper are using a realistic environment and publishing a huge dataset for the entire community. The paper also talks about evaluating benchmarks on a wide variety of algorithms. Lastly the paper also talks about using Gen AI to address the gap between real world and simulation world.

**Additional Feedback:**

No additional feedback

**Documentation:**

Yes the data is extensive

**Ethics:**

There are some ethical concerns related to data privacy.

**Limitations:**

- The paper does talk about simulated data, but integration with real data is missing in the paper
- The paper does not talk about ethical consideratinos specifically related to data privacy for advertizement.

**Opportunities For Improvement:**

- The paper has some factual inaccuracies while using Gen AI models.
- A detailed documentation with tutorials and use cases will also help

**Relation To Prior Work:**

Yes

**Summary And Contributions:**

The paper talks about Large-Scale Auction (LSA) Benchmark, designed to improve research on decision-making in competitive environments like online advertising. It includes a realistic simulation platform, the LSA Environment, and a huge dataset with 500 million records. The LSA Environment models the bidding process advertisers use to win ad placements and evaluates different strategies using advanced algorithms. Also, the paper discusses two main tasks: maximizing value within budget limits and controlling costs per action. The results show that some algorithms perform better than others, and the benchmark aims to help researchers develop better decision-making technologies.

---

> ### Author Rebuttal · Authors · 2024-08-16
>
> Thank you for your approval and positive comments.
>
> > **Opportunities For Improvement:**
> >
> > - The paper has some factual inaccuracies while using Gen AI models.
> > - A detailed documentation with tutorials and use cases will also help
>
> We will re-check our statements about Gen AI models in the revision. As for the documentation and tutorials, we plan to release these materials at a future date after we have refined the documentation.
>
> > **Limitations:**
> >
> > - The paper does talk about simulated data, but integration with real data is missing in the paper
> > - The paper does not talk about ethical consideratinos specifically related to data privacy for advertizement.
> >
> > **Ethics:** There are some ethical concerns related to data privacy.
>
> The real data is sensitive in online advertising, as it is related to the identity information of users. Therefore, we will try to provide similar generated data rather than real data. Our generative model is trained with real data and can be seen as the surrogate of real data. Moreover, we think our empirical results can be evidences of the similarity between generated data and real data to some extents. Additionally, the scale of the real data is enormous, potentially reaching hundreds of billions of records. Using a generative model is more beneficial for the platform's lightweight design compared to directly using real data.
>
> As for the ethical consideration, the features of impression opportunities contain users' private information in online advertising. To prevent privacy leaks, we first remove sensitive information such as users' names and phone numbers from the data, keeping only demographic attributes. On this basis, we use a generative model to learn the patterns of impression opportunity distribution to obtain the information we need, rather than directly using real data that contains sensitive information, which is intended to address the ethical consideration.

---

### Decision · Program_Chairs · 2024-09-26

**Decision:**

Accept (Spotlight)

**Comment:**

The paper provides a large-scale benchmark for auctions arising in online advertising. The reviewers and I are in agreement that it constitutes a significant advancement in quantifying progress in the field and will provide a useful evaluation platform, and is ready for publication. We recommend addressing a few last minor concerns for the final publication and ongoing maintenance of the benchmark, especially around additional documentation and tutorials, and potential privacy concerns.

This benchmark has the potential to decrease the gap between methods and real world applications, which is very exciting.